# Combined Use of Antimicrobial Peptides with Antiseptics against Multidrug-Resistant Bacteria: Pros and Cons

**DOI:** 10.3390/pharmaceutics15010291

**Published:** 2023-01-14

**Authors:** Maria S. Zharkova, Aleksey S. Komlev, Tatiana A. Filatenkova, Maria S. Sukhareva, Elizaveta V. Vladimirova, Andrey S. Trulioff, Dmitriy S. Orlov, Alexander V. Dmitriev, Anna G. Afinogenova, Anna A. Spiridonova, Olga V. Shamova

**Affiliations:** 1Institute of Experimental Medicine, WCRC “Center for Personalized Medicine”, 12 Academic Pavlov Street, St. Petersburg 197022, Russia; 2St. Petersburg Pasteur Institute, 14 Mira Street, St. Petersburg 197101, Russia; 3Department of Clinical Microbiology, Pavlov First Saint Petersburg State Medical University, 6/8 Lev Tolstoy Street, St. Petersburg 197022, Russia; 4Department of Biochemistry, Saint Petersburg State University, 7/9 Universitetskaya Embankment, St. Petersburg 199034, Russia

**Keywords:** antimicrobial peptides, antiseptics, combined use, drug-resistant bacteria, antibacterial action, biofilms

## Abstract

Antimicrobial peptides (AMPs) are acknowledged as a promising template for designing new antimicrobials. At the same time, existing toxicity issues and limitations in their pharmacokinetics make topical application one of the less complicated routes to put AMPs-based therapeutics into actual medical practice. Antiseptics are one of the common components for topical treatment potent against antibiotic-resistant pathogens but often with toxicity limitations of their own. Thus, the interaction of AMPs and antiseptics is an interesting topic that is also less explored than combined action of AMPs and antibiotics. Herein, we analyzed antibacterial, antibiofilm, and cytotoxic activity of combinations of both membranolytic and non-membranolytic AMPs with a number of antiseptic agents. Fractional concentration indices were used as a measure of possible effective concentration reduction achievable due to combined application. Cases of both synergistic and antagonistic interaction with certain antiseptics and surfactants were identified, and trends in the occurrence of these types of interaction were discussed. The data may be of use for AMP-based drug development and suggest that the topic requires further attention for successfully integrating AMPs-based products in the context of complex treatment. AMP/antiseptic combinations show promise for creating topical formulations with improved activity, lowered toxicity, and, presumably, decreased chances of inducing bacterial resistance. However, careful assessment is required to avoid AMP neutralization by certain antiseptic classes in either complex drug design or AMP application alongside other therapeutics/care products.

## 1. Introduction

Modern treatment and healthcare heavily rely on the use of antimicrobial agents. Classic antibiotics, which have been widely introduced into practice since the 1940s, quickly became a linchpin in the development and flourishing of many medical fields: from the obvious therapy of infectious diseases and immunodeficient conditions to any invasive surgical procedures, transplantation, use of various implants or endoprostheses, radio- or chemotherapy, and other techniques that pose the risk of or outright induce infection or immunosuppression as a side or intended effect [1,2].

Unfortunately, microorganisms, as any other organisms, tend to evolve facing selective pressure, probably better than most, regarding their duplication frequency, population size, and resulting evolution rate. Once acquired via natural mutagenesis or from other sources, resistance genes may provide their carriers with significant advantage if exposed to sublethal concentrations of antimicrobial drugs, fixating the resistance in the population. The possibility of horizontal gene transfer between different types of microorganisms allows such resistance to quickly spread among strains, including the pathogenic ones [3,4].

The ongoing antimicrobial resistance crisis is a matter of great public concern worldwide [4,5,6]. The Global Action Plan on antimicrobial resistance outlined by the World Health Organization [7] indicates two main strategic directions requiring scientific, economic, and social effort. One is to intensify and encourage the development of new effective antimicrobial drugs, combinations, or treatment strategies. The other is to establish better control over the proper use of existing and future antimicrobials to postpone resistance emergence and spread, because, as the current antibiotic development trend suggests [1], if the matter is left unchecked, that will be a race we are unlikely to win.

Antimicrobial peptides (AMPs) are considered one of the promising classes of compounds to serve as templates for designing new antimicrobial drugs and materials [8,9,10]. Most of the members of this group isolated from biological sources naturally function as effector molecules actively participating in the inactivation of microbial pathogens by the innate immune system of the host [11]. AMPs are identified throughout the tree of life, except for non-cellular life forms. They are reported in plants, fungi, animals, and bacteria [12]; however, there is a slight inconsistency as to whether only ribosomally synthesized ones with minor post-translational modifications or any peptidic antimicrobial molecules, including lipopeptides, should be labeled as AMPs [12,13]. Besides extraction from natural sources, AMPs can be produced via peptide synthesis or various recombinant expression systems that provide options for their modification and scalable manufacturing [10].

AMPs are generally cationic and amphipathic. Their main mechanism of action is based on electrostatic binding and rapid non-enzymatic damaging of microbial membranes supplemented by the possibility of further targeting intracellular substances such as negatively charged nucleic acids or some more specific interactions with certain proteins, ribosomes, and other structures, all in all sabotaging key cellular processes [14,15,16]. Relative non-specificity and multifaceted effects define AMPs’ wide spectrum of activity, including strains with multiple drug resistance, and make them a harder target in the development of highly effective resistance mechanisms [17,18,19]. They are reported to be effective against microbial biofilms [20,21,22]. As small, naturally occurring molecules they are considered to have quite a low immunogenicity and little chance of causing accumulation, metabolization, and excretion problems. Immunomodulatory and wound healing capabilities of some AMPs are regarded as an additional boon for multipurpose drug design [8]. However, the idea of adopting AMPs as therapeutics is not without problems. Beside the questions of mass-production strategy and cost, the key ones are a rather low selectivity, as AMPs often exhibit a fraction of their toxic effects against eukaryotic cells as well [23,24], and a number of the pharmacokinetics-related issues concerning the optimal administration method, targeted delivery, and the maintenance of stability in various biological fluids [8,25]. Additionally, preserving biological activity of the peptides in some dosage forms, e.g., various solutions, for prolonged periods of time may require careful conditioning. Immunomodulatory properties, especially of the organism’s own peptides, should also be approached with caution to avoid triggering autoimmune processes by uncontrolled general administration [26,27]. Although considerable progress in resolving the listed issues has already been made, both systemic reviews [25,28] and overall statistics among the AMP derivatives currently undergoing clinical trials [29,30] indicate that the most accessible area for the early introduction of AMP-based drugs is topical application. Another compromise niche is their use in creating antimicrobial coatings on medical devices and dressings [31,32,33].

In light of this fact, the question of the interaction of AMPs with various antiseptic agents and other components of disinfectants is of clear interest. Antiseptics serve as veritable medical workhorses in local treatment and infection prophylactics. They are widely used for disinfection of operating fields and of various medical devices as well as in wound care for wound cleaning, pus removal, and irrigation of the local inflammation foci [34,35,36]. Their mechanism of action is usually also less specific than that of antibiotics; they are active against many antibiotic-resistant strains, and the development of specific resistance to antiseptics themselves is considered unlikely [37]. They are also regarded as promising antibiofilm agents [38,39]. However, antiseptics often exhibit considerable toxic side effects, the reduction of which, if possible, would be a desirable outcome [40,41,42]. The toxicity of some antiseptic classes is much more prominent than comparably minor toxicity concerns regarding AMP application; they are often addressed as biocides [37] for their universal damaging effect against biological objects. Thus, the interaction between AMPs and antiseptics may turn out to be a source of both possibilities and problems.

Herein, we analyzed combined effects of a number of AMPs and antiseptic agents to evaluate the possibilities of their synergistic interaction toward drug-resistant bacteria and biofilms and to assess whether their toxic effects toward host cells were enhanced as well. Combinations can be used to more efficiently overcome microbial resistance [43,44,45], and synergy allows a reduction in the effective doses and, hence, provides a possibility to increase the selectivity of antibacterial action and to reduce toxic side effects toward the host [46].

We selected three AMPs with different modes of action (protegrin 1 (PG-1), which is membranolytic; bactenecin ChBac3.4, which combines membranolytic and intracellular mechanisms; and its structural analog, RFR-ChBac3.4(1–14), for which intracellular targets are predominant) and tested them in combinations with a set of antiseptics (sodium hypochlorite, prontosan, dioxydin, poviargolum, etidronic acid). The main idea was to verify whether there would be some type of benefit in interaction with antiseptics depending on one type of action of the AMP or another. In addition, we also tried a couple of surfactants (anionic sodium lauroyl sarcosinate and amphoteric cocamidopropyl betaine). Though not necessarily strong as individual antimicrobials, surfactants are common components of antiseptic compositions and disinfectants and, due to their similar amphiphilic nature, are interesting candidates to promote AMPs’ membranolytic properties.

## 2. Materials and Methods

### 2.1. Materials

#### 2.1.1. Antimicrobial Peptides

PG-1 was kindly provided by Prof. R. Lehrer (University of California, Los Angeles, CA, USA); it was synthesized by SynPep Corporation (Dublin, CA, USA) with assessed purity of 99%. ChBac3.4 and RFR-ChBac3.4(1–14) were produced as previously described [47] according to a standard solid-phase peptide synthesis protocol utilizing Fmoc/tBu protecting groups scheme on a Symphony X peptide synthesizer (Protein Technologies, Tucson, AZ, USA). Trifluoroacetic acid (TFA) cleavage cocktail (TFA/triisopropylsilane/water/ethandithiol = 94/1/2.5/2.5) was used for the final deprotection and cleavage of the assembled linear peptides from the 2-chlorotrityl chloride resin. Samples were purified via semi-preparative RP-HPLC using a Waters SymmetryPrep C18 column, 9 × 300 mm, 100Å, 7µm, and then verified for a purity of no less than 96% by analytical RP-HPLC using Luna C18 column, 4.6 × 250 mm, 100 Å, 5 µm on a Gilson chromatograph. The molecular weight of the peptides was checked to be as expected by MALDI-TOF mass spectrometry with alpha-cyano-4-hydroxycinnamic acid as a matrix.

#### 2.1.2. Antiseptics and Surfactants

Prontosan wound irrigation solution (B. Braun; Melsungen, Germany), poviargolum powder (FSUE “SCTB ‘Technolog’”; Saint Petersburg, Russia), Xydifon (commercial name of etidronic acid 20% aqueous solution; OJSC “MosChemPharm” named after N.A. Semashko; Moscow, Russia), and Dioxydin 1% aqueous solution (OJSC “MosChemPharm” named after N.A. Semashko; Moscow, Russia) were purchased from the market. CP grade powders of sodium lauroyl sarcosinate (Sigma-Aldrich, St. Louis, MO, USA) and cocamidopropyl betaine (Evonik Industries, Essen, Germany) were helpfully provided by our colleagues from the Institute of Macromolecular Compounds of RAS and CP grade sodium hypochlorite 10% aqueous solution by our colleagues from the Faculty of Dentistry and Medical Technology of the Saint Petersburg State University.

#### 2.1.3. Bacterial Strains

Multidrug-resistant bacterial isolates were initially obtained from infected wounds (or from urine in the case of *Escherichia coli* ESBL 521/17) and originated from the collections of the Research Institute of Epidemiology and Microbiology named after L. Pasteur, Saint Petersburg, Russia and of the Saint Petersburg State University. Antimicrobial resistance spectra of all five used clinical isolates were listed in detail—with determined minimal inhibitory concentrations (MICs) for antibiotics—previously in the supplementary data of [47]. Briefly, *E. coli* ESBL 521/17 showed resistance to ampicillin, cefotaxime, ceftazidime, cefixime, aztreonam, netilmicin, ciprofloxacin, erythromycin, and trimethoprim/sulfamethoxazole and amoxicillin/clavulonic acid combinations; *Acinetobacter baumannii* 7226/16 was resistant to imipenem, gentamicin, tobramycin, ciprofloxacin, erythromycin, and trimethoprim/sulfamethoxazole combination; *Pseudomonas aeruginosa* MDR 522/17 was found to be resistant to meropenem, ceftazidime, cefixime, gentamycin, netilmicin, ciprofloxacin, erythromycin, and colistin; *Klebsiella pneumoniae* ESBL 344/17 was resistant to ampicillin and erythromycin; *Staphylococcus aureus* 1399/17 demonstrated resistance to the action of ampicillin, oxacillin, gentamicin, amikacin, ofloxacin, and erythromycin. *Escherichia coli* ML-35p laboratory strain was a generous gift from Prof. R. Lehrer (University of California, Los Angeles, CA, USA).

### 2.2. Methods for Examining Antibacterial Effects

#### 2.2.1. Broth Microdilution Assay: Evaluation of Minimal Inhibitory Concentrations against Planktonic Bacteria

Antimicrobial activity of the test substances against planktonic bacteria was measured according to the standard broth microdilution assay protocol suggested by the guidelines of the European Committee for Antimicrobial Susceptibility Testing with subtle adjustments proposed previously to minimize the non-specific binding of AMPs to the surface of the microplates [48,49]. In this regard the test plates were preincubated for 1 h at 37 °C with 0.1% bovine serum albumin (BSA) in distilled water before use. Bacteria were grown overnight at 37 °C with shaking in 2.1% Müller–Hinton broth (MHB), and then a small aliquot was transferred into the fresh portion of the same medium for another 2–3 h on the day of the experiment to provide us with the log phase bacterial suspension. Concentration of bacterial cells in suspension was evaluated based on absorbance at 620 nm and diluted down to 1 × 10^6^ CFU/mL with MHB. Equal volumes of the prepared bacterial suspension in MHB and two-fold serial dilutions of the antimicrobials in 10 mM sodium phosphate (Na-P) buffer (pH 7.4) with 0.1% BSA were mixed and incubated overnight at 37 °C. The lowest concentration where no bacterial growth was observed the next day under visual examination was considered the minimal inhibitory concentration. The test was repeated at least three times, and the final MIC value was established as a median result. Each experiment was conducted in triplicates. Terasaki microplates (10 µL end volume, V-shaped bottom; Sarstedt, Nümbrecht, Germany) were used.

#### 2.2.2. Checkerboard Titration for Analyzing Combined Antibacterial Effect against Planktonic Bacteria

The mode of interaction of the test substances while applied in combinations was analyzed based on the calculation of the fractional (or combination) indices—the mathematical “embodiment” of the Loewe additivity paradigm [50,51] that we used for the classification of combination effects in this study. As synergy is generally defined as a greater than additive effect, and antagonism as a lower than additive effect, the main problem is to establish a reference level for this additive effect [52]. The Loewe approach uses the idea of the self-additivity of an individual substance; thus, the individual activity may be regarded as an additivity control. To identify additivity, synergy, or antagonism in Loewe terms, one should choose a fixed level of the studied effect and then compare the doses of individual substances required to reach this level, with the fractional doses of the substances in combination providing the same level of the effect.

In the case of activity testing of combinations toward planktonic bacteria, a checkerboard titration [53,54] scheme was used to create an array of mixtures of the two substances (A and B) in various concentrations by diluting the first one two-fold row-by-row and the second one column-by-column. The left column and the bottom row contained the dilutions of only one substance (A or B) to control their individual activity levels in every experiment. Aside from the antimicrobial part being a mixture, the experiment followed the microdilution assay protocol. Once the wells of the test plate where the bacterial growth was completely inhibited by the mixture of the components A and B were identified, fractional inhibitory concentration indices (FICIs) were calculated for them as FICI = [A]/[MIC A] + [B]/[MIC B], where [A] and [B] are concentrations of A and B in the bacterial-growth-free well, and [MIC A] and [MIC B] are MICs of A or B illustrating their individual antimicrobial potency. Based on the minimal obtained value of FICI, A and B interaction was classified as synergy (minFICI ≤ 0.5; effects are amplified), additivity (0.5 < minFICI ≤ 1; effects sum up), independent action (1 < minFICI ≤ 2; effects stay the same as if each component acts alone), or antagonism (minFICI > 2; effects are subdued); thresholds are chosen taking into consideration the two-fold serial dilution error margin [53,54,55].

#### 2.2.3. Bacterial Membrane Permeability Assays

Monitoring the ability of the test substances or their combinations to affect bacterial membranes and enhance their permeability over time was organized using a method first proposed by Prof. Lehrer’s scientific group [56]. The idea is based on utilizing special markers that cannot spontaneously penetrate bacterial membranes until those are damaged but would be readily processed by the bacteria’s own enzymes into products of different color once they enter. Then, the product can be spectrophotometrically detected, and the dynamics of its accumulation can be associated with the scale and swiftness of the inflicted membrane impairment. The specific bacterial strain *E. coli* ML-35p was designed for the procedure. It is a descendant of *E. coli* ML-35, first identified during *lac* operon studies, deficient in lactopermease but constitutively expressing cytoplasmic β-galactosidase [57], that was modified with a plasmid vector to additionally provide it with a periplasmic β-lactamase. These modifications create the basis for the mentioned membrane damage detection approach, as there are two specifically placed enzymes, but their lactose/galactose-based substrates cannot enter bacterial cells on their own in the absence of lactopermeases.

Nitrocefin (Calbiochem-Novabiochem, San Diego, CA, USA), which can be converted by β-lactamase into a pink-colored product detectable at 486 nm once it reaches the periplasmic space, was used as a marker of the outer membrane permeability, and ONPG (*o*-nitrophenyl-β-D-galactoside, Sigma, St. Louis, MO, USA), producing yellow-colored *o*-nitrophenyl detectable at 420 nm once hydrolyzed by β-galactosidase in the bacterial cytoplasm, was likewise used as an indicator of inner membrane damage.

An overnight culture of *E. coli* ML-35p grown at 37 °C in 3% Trypticase soy broth (TSB) was washed of the medium twice with 10 mM Na-P buffer (pH 7.4) via a 10 min long centrifugation at 600 g and 4 °C and diluted down in the same buffer to 1 × 10^8^ CFU/mL stock concentration based on the optical density (OD) of the suspension at 620 nm (the formula being concentration [CFU/mL] = OD_620_ × 2.5 × 10^8^). The stock was preferably used immediately or kept at 4ᵒC for no more than 1 h. Experiments were carried out in 96-well clear flat-bottom plates. The 100 µL samples containing antimicrobials or their combinations in desired concentrations, 10 mM Na-P buffer (pH 7.4), 100 mM NaCl, 20 μM of nitrocefin or 2.5 mM of ONPG, and bacteria in a final concentration of 2.5 × 10^7^ CFU/mL were made in duplicates. Controls included no antimicrobial substances. Bacterial suspension was the last component and was added just before the start of the measurement. OD values at the wavelengths of 420 and 486 nm were collected once per minute for 1–2 h using SpectraMax 250 Microplate Spectrophotometer (Molecular Devices, San Jose, CA, USA); samples were maintained at 37 °C with 5 s of shaking before each read.

#### 2.2.4. Fluorometric Resazurin Dye-Based Assay to Monitor Bacterial Metabolic Activity and Viability

The decline in metabolic activity of bacterial cells faced against test substances or their combinations was tracked using resazurin dye, also known as alamarBlue. Due to resazurin’s ability to be irreversibly reduced to fluorescent pink-colored resorufin in actively metabolizing cells, it is widely used for testing viability, proliferation, metabolic activity, respiration, etc., in various types of cells [58]. As it was initially suggested by the investigation of the biochemical background of the resazurin reduction conducted in spermatozoa, this process is promoted by NADH + H^+^ and is dependent on the activity of dehydrogenases (diaphorases) of the electron transport chain [59]. Thus, resazurin-to-resorufin conversion was primarily associated with energy metabolism impairment [60]. However, resazurin’s oxidation–reduction potential indicates that it can accept electrons from other molecules (NADPH, FADH, FMNH [flavin mononucleotide hydride], and cytochromes), and its transformation is not strictly specific to the electron chain disruption [58].

The experimental procedure was basically the same as for permeability testing except for the substitution of the markers by 120 μM resazurin and the addition of 0.1% MHB to provide bacteria with a metabolic substrate. An additional control without both antimicrobials and bacterial cells was included to imitate the zero metabolic activity level. Though resazurin-to-resorufin transformation can be observed via either spectrometry or fluorimetry, the latter was preferred as a more sensitive method. Experiments were conducted in 96-well white opaque plates. Temperature and shaking parameters were left unchanged. Fluorescence intensity was measured at 590 nm with excitation at 560 nm. Data were collected every 3 min for 4–6 h using a Gemini EM plate spectrofluorimeter (Molecular Devices, San Jose, CA, USA). 

#### 2.2.5. Crystal Violet Assay to Evaluate Formation of Biofilms

The impact of the tested antimicrobials and their combinations on the formation of biofilms was quantitated as per the crystal violet assay protocol published by Merritt et al. [61]. Testing was performed in 96-well immunological plates with U-shaped bottoms. Overnight cultures of tested bacteria in MHB were diluted by a factor of 50, mixed 1:1 with 50 μL of active compounds or combinations in the same growth medium adjusted to double the desired final concentration, and left to form biofilms for 24 h at 37 °C. Experimental samples were quadruplicated, and control without antimicrobials was octuplicated.

Then, the liquid content of the test plates was discarded, and the plates were gently washed in still water from residual traces of unattached planktonic bacteria. Each well was supplied with 125 μL 0.1% aqueous solution of crystal violet dye for 10 min at room temperature to stain biofilm components left attached to the surface. After staining, the plates were washed with clear water once more to remove excessive dye and left to air-dry. To quantify the amount of the bound dye corresponding to the thickness of the formed biofilm, it was redissolved in 30% acetic acid, 200 μL of which was introduced into each well. After 15 min at room temperature, the content of the wells was mixed by repetitive pipetting to ensure complete dye extraction, and 125 μL of each sample was transferred well-by-well into 96-well clear flat-bottom plates for spectrophotometric examination. OD was measured at 580 nm. Final results were calculated as medians based on three independent experiments.

### 2.3. Methods for Examining Toxicity toward Eukaryotic Cells

#### 2.3.1. Hemolysis Test 

Hemolytic properties of the test substances and their combinations were assessed as previously described [62]. Whole human blood was collected from healthy donors by venipuncture into sterile EDTA-covered vacuum tubes, diluted 10 times with ice-cold phosphate-buffered saline (PBS) containing 4mM EDTA for the start, and then washed out twice with still ice-cold but anticoagulant-free PBS by a 10 min long centrifugation at 300× *g* and 4 °C to remove plasma and anticoagulant altogether. The resulting cell precipitate was assigned as 100% red blood cells (RBCs) and used to prepare a 3.1% *v*/*v* stock suspension in PBS that was maintained at 4 °C before testing for no more than 3–4 days. Six µL portions of the test substances or their combinations diluted in PBS down to five times the various desired final concentrations were added to 0.5 mL microtubes in triplicates and mixed with 24 µL of 3.1% *v*/*v* RBC suspension (that provided a 2.5% *v*/*v* RBC final concentration). Samples were incubated for 30 min at 37 °C and then supplied with 90 µL of ice-cold PBS to stop further hemolysis. Remaining RBCs or their debris were precipitated by 4 min of centrifugation at 10,000× *g* and room temperature, and 100 µL of the supernatants was transferred to a 96-well clear flat-bottom test plate to spectrophotometrically detect hemoglobin released from the damaged RBCs. OD was measured at 540 nm—one of the hemoglobin absorbance maxima. The samples with 6 µL of pure PBS (with no potentially hemolytic agents) were used as a 0% lysis reference point. Experimental samples were compared with 0% lysis control by the Mann–Whitney U-test (*p* < 0.05) to identify statistically significant hemolytic action. In case the exact level of effect was of interest, the samples with 6 µL of 5% *v*/*v* Triton X-100 in PBS were also prepared to additionally obtain a 100% lysis reference point and calculate the percentage of hemolysis in test samples as hemolysis (%) = (OD_sample_ − OD_0% lysis_)/(OD_100% lysis_ − OD_0% lysis_) × 100%. At least three independent repeats were performed.

#### 2.3.2. MTT Test

Potential cytotoxic effects of the test substances and their combinations toward human dermal fibroblasts were observed using a MTT dye [3-(4,5-dimethylthiazol-2-yl)-2,5-diphenyltetrazolium bromide]-based test first proposed by Mosmann [63]. MTT basically functions on the same principle as resazurin: it is reduced in metabolically active cells to the product named formazan, which differs from MTT by its properties. While MTT is yellow, formazan is blue and water-insoluble. Normal human fibroblasts initially obtained from the Pokrovsky Stem Cell Bank (Saint Petersburg, Russia) were cultured at 37 °C and 5% CO_2_ in DMEM medium supplemented with 10% fetal bovine serum, glutamine, and Pen-Strep. The day before, experiment cells were detached from the culture flasks using Trypsin-Versene reagent, collected into a sterile 15 mL tube, centrifuged for 10 min at 300× *g* to discard the previous medium, and resuspended in serum-, glutamine-, and antibiotic-free DMEM. Cells were counted using a hemocytometer; their concentration was adjusted, and they were distributed at 10,000–15,000 cells per well in 90 µL in a 96-well sterile treated flat-bottom plate and left at 37 °C and 5% CO_2_ overnight to properly adhere. The following day, their well-being was visually checked using an inverted light microscope, and the wells were supplied with 10 µL of the test substances or their combinations in DMEM in 10 times the final desired concentrations. Experimental samples were made in triplicates. Ten µL of DMEM with no active compounds was added to the cell-containing wells to create a 100% viability control, and the wells with 100 µL of sterile DMEM were used as a 0% viability control. At least six repeats of each type of the control samples were made. Test plates were incubated at 37 °C and 5% CO_2_ overnight, and 10 µL of the 5 mg/mL MTT in PBS were introduced into each well the next morning. After another 4 h of incubation at the same conditions, formed formazan crystals were dissolved by adding 110 µL of isopropanol containing 0.04 M HCl (the latter is used to “switch” phenol red pH indicator in DMEM from red to yellow). Then, each sample was thoroughly mixed by repetitive pipetting, and formazan concentration was spectrophotometrically evaluated measuring OD at 540 nm with subtraction of the background at 690 nm. Similar to the hemolysis, statistical significance of the effect was determined by comparing experimental samples with 100% viability control (Mann–Whitney U-test; *p* < 0.05), and the effect’s magnitude was calculated as viable cells (%) = (OD_sample_ − OD_0% viability_)/(OD_100% viability_ − OD_0% viability_) × 100%.

#### 2.3.3. Assessment of Viability of Adherent Cells by Light Microscopy

We resorted to the direct cell count in case the MTT test was not applicable or required verification of some sort. This idea was borrowed from the crystal violet assay for determining viability of cultured cells [64] and is based on the fact that, once dead, adherent cells detach from the surface. Since we were not aiming to analyze a large number of samples in this manner, we simply skipped the staining step of the crystal violet assay, which is also slightly difficult and brings a risk of detaching cells during washing. We proceeded with the same protocol as for the MTT test down to the step of adding MTT dye. Then, instead, we used an inverted light microscope (magnification ×500) to calculate the number of attached cells in 10 fields of view in each experimental well of interest and compared it with the count for the 100% viability control wells. The exact effect was evaluated as viable cells (%) = mean number of adhered cells in a sample / mean number of adhered cells in 100% viability control × 100%.

### 2.4. General Principles Used for Combined Effect Analysis

While analyzing time–effect curves, the general idea was to proceed with a combination containing each component in ¼ of its minimal effective concentration (MEC or MIC) that corresponded to a fractional index of 0.5—the threshold value for synergy. The effect of a combination was then compared with individual effects of the components in concentrations equal to ¼ or ½ of their MECs or MICs. The higher individual concentration was to some extent regarded as a reference for additive effect, as, according to the Loewe additivity paradigm [50] that we used to classify various types of combined effects in our study, the compounds must be self-additive (and ¼ + ¼ = ½).

For the end point effects, we consistently used the same fractional indices-based approach (as FICI for antimicrobial activity). First, the effective concentrations of the individual substances providing a certain level of effect were established, usually by comparing to a reference effect level provided by control samples. Then, the mixtures of the compounds in a variety of concentrations were tested for the same level of effect, and fractional indices, basically characterizing the reduction in the effective concentrations of the components that could be achieved by combined use, were analyzed. In case we were more interested in verifying the absence of the synergistic effect rather than in evaluating the exact magnitude of said synergy (e.g., for combined hemolytic or other undesirable toxic effects), we resorted to the use of the minimum of concentration combinations for a mixture of substances A and B:

(½ MEC A + ½ MEC B), which can result in a fractional index of 1—an upper threshold indicating additivity; (¼ MEC A + ¼ MEC B), which can result in a fractional index of 0.5—an upper threshold for synergy.

MECs (or MICs) of the individual substances due to possible fluctuation were controlled alongside the activity testing for combinations: the range from 1 dilution above down to 1–3 dilutions below the expected MEC was reobserved for the compounds every time while assessing their combined use.

## 3. Results

### 3.1. Preliminary Overview of the Tested Compounds

First, we summarized the basics of antimicrobial action of the tested compounds to discuss the results in a more organized and substantial manner.

PG-1, initially isolated from porcine leukocytes, is a short, highly active β-hairpin peptide executing its antibacterial effect via a swift membranolytic mechanism considered classic for AMPs [65,66,67]. It is shown to form toroidal pores in membrane bilayers [68,69]. Although its broad antimicrobial spectrum and steep killing curve made it an attractive candidate for practical use [29,30], native PG-1 demonstrates significant toxicity toward eukaryotic cells as well, including hemolytic effects [70,71].

Caprine ChBac3.4 is a somewhat unconventional representative of the proline-rich family of AMPs (PrAMPs) [72]. Unlike PG-1, these linear peptides are known for their low toxicity, which can be attributed to their dual mechanism of action; unless the concentration becomes quite high to trigger non-specific damage of bacterial membranes, they exploit a non-lytic mechanism, targeting peptide synthesis [15,72,73]. The blockage of the exit tunnel of bacterial ribosome and the binding of the bacterial chaperon DnaK essential for proper protein folding are among the identified intracellular strategies implemented by PrAMPs [15,74,75]. ChBac3.4’s peculiarity is that it shows distinct membranolytic abilities even around its MIC that, in turn, presumably provide it with a broader spectrum—against both Gram-positive and Gram-negative bacterial species—compared with most of its PrAMPs “brethren”, which are active predominantly against Gram-negative bacteria that have certain transporters (such as SbmA [73] and MdtM [74]), facilitating the entrance of non-membranolytic peptides into the cell.

RFR-ChBac3.4-(1-14) is a shortened 14-residue N-terminal fragment of ChBac3.4 whose diminished antimicrobial activity is compensated by an additional RFR-motif back to a level compatible with the full-length peptide. Contrary to the initial ChBac3.4, this shortened peptide shows no membranolytic effect on bacterial cytoplasmic membrane [47].

Sodium hypochlorite is quite extensively used in medicine as an antiseptic compound for minor wound treatment, pre-surgical disinfection, or some more specific cases, such as root channel irrigation in endodontics [76,77]. In both hypochlorite ions ^−^OCl and hypochlorous acid HOCl, present in aqueous solution, chlorine exists as Cl^+^, lending its own electron to the more electronegative oxygen. This makes Cl+ a strong electrophile, searching for a spare pair of electrons, that readily oxidizes various biological molecules, attacking thiol and amino groups and even peptide or C=C bonds, which obviously interferes with these molecules’ functioning [77,78,79].

Dioxydin (also spelled dioxidine or dioxidin in different publications) is a derivative of the di-N-oxide of quinoxaline that inhibits DNA synthesis and disrupts nucleic metabolism [80,81,82]. It has been successfully used in topical treatment of pyogenic infections for over 30 years. It provides a bactericidal effect and has a broad spectrum of activity, including drug-resistant bacterial strains. Its action on DNA is detectable even in sub-MIC levels and is believed to intensify under anaerobic conditions due to the promotion of reactive oxygen species creation [83,84].

Poviargolum is a colloidal preparation of metallic silver stabilized with polyvinylpyrrolidone. It is designed for the treatment of pyogenic infections in various areas, from dermatology and traumatology to ophthalmology and dentistry [85]. Its antimicrobial effect seems to be primarily mediated via the release of silver ions. Though the entirety of the mechanism of action of both ionic and nano forms of silver is yet to be understood in all its complexity, one of the main features is believed to be electrostatic interaction with various negatively charged compounds. Within the bacterial envelope, it leads to membrane destabilization, depolarization, and dissipation of transmembrane potential; in the case of nucleic acids, it results, for example, in the condensation of DNA, which prevents proper transcription; in the case of phosphate groups of various proteins acting like switches in different signaling pathways due to timely phosphorylation/dephosphorylation, it blocks such transduction. Another key factor in silver’s effect against bacteria is interference with thiol, amino, and other groups with high electron density, which impacts greatly on the functioning of various enzymes and structural proteins [86,87,88].

Prontosan wound irrigation solution as produced by the manufacturer is used for the treatment of various acute, chronic, and burn wounds [89]. The composition includes 0.1% polyhexanide and 0.1% undecylenamidopropyl betaine. The former is a broad-spectrum polycationic antimicrobial that has been used as an antiseptic or disinfectant for several decades without reported bacterial resistance [89,90], and the latter is an amphoteric surfactant facilitating the removal of bacterial biofilms and wound debris [89]. The data on the mechanism of antimicrobial action of polyhexanide are somewhat controversial. Earlier, it was assumed to affect membrane permeability with some similarity to AMPs [91,92]. Recent works, however, suggest that polyhexanide translocates through the membrane without causing substantial damage [90]. Thus, its reported abilities to bind proteins and DNA, interfering with transcription and replication, should play a major role in its antimicrobial effect [90,92,93].

Etidronic acid is a bisphosphonate compound. Although bisphosphonates in medical practice are primarily used in the treatment of the conditions associated with metabolic disorders in bone tissue [94], the possibility of utilizing etidronic acid as an antiseptic for prevention and treatment of inflammatory diseases of the oral cavity was also confirmed in laboratory and clinical studies [95]. On top of that, it was revealed that etidronic acid has the ability to inhibit bacterial β-lactamases, including metallo-β-lactamase [96]. The basis of its antimicrobial effect, however, remain unclear. Some redox, pH-related, or even Ca^2+^-binding events [94] may play a role in altering vital processes in bacterial cells.

Surfactants, as aforementioned, are considered to be rather weak antimicrobials by themselves. Their effect is based on the ability to thoroughly disrupt lipid membranes (as in the “carpet” model proposed for AMPs, which is also referred to as “detergent-like”), the same that is put to good use for extracting membrane proteins [97,98]. While antimicrobial capabilities of the anionic surfactant sodium lauroyl sarcosinate, also known as sarcosyl, are acknowledged [99], the general notion on amphoteric amidopropyl betaines is that they have low antibacterial efficiency [100,101]. Aside from the antimicrobial activity, we also should mention that cocamidopropyl betaine (the main component of which is lauramidopropyl betaine), which was tested as an individual agent, and undecylenamidopropyl betaine, which is a prontosan component, are chemically close compounds. Lauric (dodecanoic) acid has a chain of 12 carbon atoms, and undecylenic acid has a chain of 11 carbon atoms with a double bond between the 10th and the 11th ones.

### 3.2. Antimicrobial Action of the Individual Compounds against Planktonic Bacteria

Antibacterial activity was tested against five clinically isolated bacteria (four Gram-negative and one Gram-positive) with multiple drug resistance to commonly used antibiotics and also against a specific laboratory strain, *E. coli* ML-35p, frequently utilized in analyzing the effect of AMPs or other substances on the permeability of bacterial membranes. The minimum inhibitory concentrations (MIC) characterizing the individual antibacterial effects of AMPs, antiseptic agents, or surfactants of interest toward these bacteria in their planktonic state are given in Table 1.

All drugs, antimicrobial peptides, and antiseptics alike—and, to a lesser extent, surfactants—show activity against multidrug-resistant bacteria within the expected concentration range. For peptides, the effect toward the selected bacterial isolates was already discussed earlier alongside the prospects of their combined action with conventional antibiotics [47,62]. Among the antiseptics, the smallest activity was demonstrated by etidronic acid: toward a number of tested bacteria, the established MIC values turned out to be higher than the recommended therapeutic concentration of 2% (20 mg/mL) [95]. At the same time, it is interesting to note that the MICs that were the closest to the recommended dose were found against the bacterial stains *A. baumannii* 7226/16 and *P. aeruginosa* MDR 522/17, known to actively form biofilms.

Sodium hypochlorite exhibits its bactericidal effect within the range of concentrations usually used for medical purposes, from 0.5% (5 mg/mL) sodium hypochlorite, known as Dakin solution, which was first utilized as an antiseptic agent for the treatment of infected wounds during World War I, up to 2.5% sodium hypochlorite, proposed for decontamination of operating fields, with the highest referenced concentration being 5.25% sodium hypochlorite, tested for irrigation of root canals in endodontics [76].

Dioxydin seems to be noticeably more effective against Gram-negative bacteria than against the Gram-positive *S. aureus* 1399/17. Considering its activity toward a number of *S. epidermidis* bacterial isolates resistant to fluoroquinolons (yet unpublished data not included into the current study), MICs of this antiseptic required to inactivate Gram-positive bacteria are around an order of magnitude higher. The fact that all of the mentioned Gram-positive bacteria are resistant to fluoroquinolones and that dioxydin, as a derivative of quinoxaline, has a certain level of similarity to this group of antibiotics may be of note. However, no evidence of any sort of cross-resistance between dioxydin and fluoroquinolones was found in the literature; moreover, the use of dioxydin is recommended in case of resistance to other antibiotics as per its drug label [102]. Besides, the difference in susceptibility to fluoroquinolones among Gram-negative bacteria (only *K. pneumoniae* ESBL 344/17 and *E. coli* ML-35p are highly susceptible to fluoroquinolones [62]) does not affect dioxydin’s effectiveness against them. Regardless, the inhibitory concentrations of dioxydin toward both Gram-negative and Gram-positive bacteria remain no more than the 0.1–1% recommended in the manufacturer’s prescription label (1–10 mg/mL).

To a lesser extent, the shift in activity toward Gram-negative bacteria in preference to Gram-positive was also observed for poviargolum, which corresponded to the existing notions about the antimicrobial capacities of nano-forms of silver. Literature data seem to indicate a greater activity of silver nanoparticles against Gram-negative bacterial species than against Gram-positive ones [103,104]. Our results considering individual antibacterial action of poviargolum are in good agreement with the data of clinical trials previously conducted by the Research Institute of Traumatology and Orthopedics (Saratov, Russia) and published by the poviargolum manufacturer [85]. According to the trials, the minimum bactericidal concentrations of poviargolum against a wide palette of bacteria isolated from purulent wounds were no more than 100–200 μg/mL.

Unexpectedly, compared to other clinical isolates, the strain *A. baumannii* 7226/16 showed markedly increased resistance to the polyhexanide (polyaminopropyl *biguanide*)-based antiseptic prontosan, but the MICs of prontosan toward all tested bacteria were, nevertheless, significantly lower than 0.1% (1 mg/mL)—the concentration of polyhexanide in a commercial prontosan wound irrigation solution.

Among surfactants, as was expected based on the literature data, sodium lauroyl sarcosinate was the most universally active toward bacteria. The obtained MIC values were overall consistent with literature data; for example, the disk-diffusion method suggests that sarcosyl MICs fall within the range of 37.5–300 mg/mL [99]. At the same time, quite peculiarly, *A. baumannii* 7226/16 and *K. pneumoniae* ESBL 344/17 demonstrated high sensitivity to cocamidopropyl betaine. This is especially interesting in light of the fact that prontosan has an amidopopyl betaine in its composition alongside the polyhexanide, and its unusually lowered MIC against *A. baumannii* is comparable with the identified individual MIC of cocamidopopyl betaine.

In summary, it can be noted that, regardless of the susceptibility profiles of the tested bacterial strains toward conventional antibiotics, both AMP and antiseptics remained effective against them.

### 3.3. Effects of Combinations against Planktonic Bacteria

Analyzing the antibacterial effect of combinations of AMPs with antiseptics and surfactants, we identified two cases of a sustainable antagonistic interaction: with sodium hypochlorite and with sodium lauroyl sarcosinate; therefore, these substances were excluded from further consideration, and the data on their action are not given in the remaining tables. The combined use of AMPs with either of the named components resulted in the enhancement of the concentrations required to suppress the growth of most of the tested bacteria by a factor of four or greater. The effect was more pronounced on the peptide side, possibly due to the concentration ratio, since, in absolute values, tested concentrations of sodium hypochlorite, and even more so of sodium lauroyl sarcosinate, were significantly higher than that of the peptides (~mg/mL for the antiseptic and surfactant vs. ~µg/mL for the peptides). A quite plausible explanation of the observed antagonism may be that the high concentration of negatively charged structures (such as the surfactant molecules and micelles that they form or hypochlorite anions) can interfere with the initial binding of positively charged AMP molecules to the negatively charged components of bacterial membranes. Ending up within the surfactant micelles, AMP molecules can be completely excluded from the active pool. In the case of sodium hypochlorite, it is also likely that the antiseptic can inflict direct damage to AMP molecules upon contact, due to their peptidic nature. 

The interaction of AMPs with other compounds selected for this study is denoted mainly as additivity or synergy. The minimal FICI for the combinations are shown in Table 2. The values interpreted as synergy are marked in bold. The most widely and universally occurring cases of synergistic interaction were found in combinations with the colloidal-silver-based antiseptic poviargolum and with the amphoteric surfactant cocamidopropyl betaine. Identified synergy with poviargolum is consistent with previous studies on antibacterial action of combinations of AMPs with silver nanoparticles [62,105,106], as well as of their conjugates [107,108]. Mutual amplification of the effects of AMPs and a surfactant may quite possibly be driven by the increase in membranolytic activity, although the effect was also observed in combination with the truncated peptide RFR-ChBac3.4(1–14), which inflicted little direct damage toward bacterial membranes. Betaine-based surfactants are known to have positive molecular charge regardless of pH (the reason that their classification as amphoteric surfactants can be consider wrong) [109]. Thus, the synergistic effect of amidopropyl betaine with AMPs could be of a similar nature to the synergy that cationic amphipathic AMPs can have with each other [110,111,112].

Combined action of dioxydin with ChBac3.4 toward *A. baumannii* and *K. pneumoniae* and with its shortened fragment RFR-ChHBAC3.4 (1-14) toward the remaining three antibiotic-resistant bacterial isolates were also identified as synergy. Prontosan showed a synergistic effect mainly with RFR-ChBac3.4(1–14)—against *A. baumannii*, *P. aeruginosa*, and *K. pneumoniae*—but also with PG-1 against *E. coli* ML-35p and with ChBac3.4 against *S. aureus* 1399/17.

Etidronic acid, due to having rather low individual activity, was tested in combinations with PG-1 and ChBac3.4. Synergy with both AMPs was found against the laboratory and clinically isolated strains of *E. coli* and against *S. aureus* 1399/17; in combination with ChBac3.4, it also acted synergistically against *A. baumannii* 7226/16. Notably, the AMPs tested were previously reported to have synergistic antimicrobial effects with β-lactam antibiotics, in particular toward bacterial strains resistant to the latter [47,62]. Moreover, the cases of synergistic interaction of AMPs and β-lactam antibiotics are quite widely described in the literature [113,114,115]. In this regard, considering triple combinations that include AMP, antibiotic, and etidronic acid that can inhibit bacterial β-lactamases may have some additional prospects.

### 3.4. Effects of the Individual Compounds and Their Combinations on the Permeability of Bacterial Membranes

Trying to take a brief glance at the possible source of the detected synergy cases between AMPs and antiseptics, we started with assessing the changes in membranolytic capabilities of the compounds used in combinations, as the disruption of bacterial membranes is the fastest killing scenario performed by most AMPs capable of it.

Here and in other cases where kinetics were compared, the curves illustrating the action of the mixture containing ¼ MIC of both tested substances (and corresponding to the FICI value of 0.5) were analyzed against the curves showing the effect of ¼ and ½ MIC of each substance individually, the latter (½ MIC) being used as a reference level of self-additivity.

The kinetic curves presented in Figure 1A demonstrate that, in the tested concentrations, dioxydin and prontosan did not significantly increase the permeability of bacterial membranes for chromogenic markers by themselves and also did not cause significant changes in the dynamics of the membranolytic action of AMPs. This supports recent notions on polyhexanide (the main antimicrobial component of prontosan) being translocated through the bacterial membrane without significantly altering its integrity by itself [90]. At the same time, our previous assumption regarding the interaction of AMPs with amidopropyl betaine was confirmed. The presence of AMPs, even of RFR-ChBAC3.4 (1-14), which does not show any substantial membranolytic activity by itself, significantly accelerated the damaging effect of the betaine surfactant on the cytoplasmic membrane of the bacteria, even in comparison with ½ MIC of betaine used alone.

Unfortunately, the chosen method for assessing the permeability of bacterial membranes using chromogenic markers and the bacteria’s own enzymes that split them turned out to be of little use for evaluating the AMP’s interaction with poviargolum or etidronic acid (Figure 1B). On the other hand, it allowed us to detect certain features regarding the individual action of these antiseptics.

Poviargolum, just like dioxydin or prontosan, exerts no significant effect on the membranolytic activity of AMPs toward the outer membrane of *E. coli* ML-35p (data not shown). Etidronic acid, as expected due to its ability to inhibit bacterial β-lactamase, prevents said enzyme from splitting nitrocefin, which is the marker used to assess the permeability of the outer membrane. However, while studying the permeability of the inner membrane, it was found that etidronic acid also fully blocked the processing of ONPG by cytoplasmic β-galactosidase, and poviargolum significantly slowed this reaction down.

The scenario of β-galactosidase being inhibited by etidronic acid or of this enzyme structure being affected by silver seems to be a fairly possible outcome based on the available data on their action. Conversely, the uncovered ability of the amidopropyl betaine surfactant to block the degradation of nitrocephin by β-lactamase was quite unexpected (Figure 1B). Since we did not conduct a more detailed analysis and found no clear evidence in the literature, we refrain from concluding whether the observed phenomenon is a direct effect on the bacterial enzyme or is of some different nature.

To verify that the observed effects of poviargolum, etidronic acid, and amidopropyl betaine were indeed associated with the reaction of the bacterial enzyme-dependent marker degradation, the curves obtained using the intact bacteria were compared with the ones obtained using the bacteria lysed by the exposure to 4×MIC of PG-1 beforehand (Figure 1B). The trends illustrating the conversion of the markers (or its absence) in the presence of the test compounds turned out to be similar in both cases.

Overall, the lack of influence on the membranolytic activity of AMPs toward bacteria discovered for most antiseptics, although not actually revealing in terms of deciphering mechanisms underlying synergy, looks promising regarding the prospects of reducing toxic side effects by the use of combinations, since the cytotoxicity of AMPs toward eukaryotic cells, the hemolytic effect in particular, is also associated with their membranolytic capabilities.

### 3.5. Effects of the Individual Compounds and Their Combinations on the Metabolic Activity of Bacteria

To have at least some insight into the interaction dynamics of AMPs with antiseptics that were not found to promote AMPs’ membranolytic action, we analyzed how they affected the bacterial metabolism at the early stages, as the microdilution method evaluates the antibacterial effect of the drugs only 18–20 h after the start of exposure. We used a fluorimetric technique to monitor the metabolic activity of bacteria utilizing resazurin dye, the reduction of which to a fluorescent product is associated with the intensity of aerobic metabolism of the living cells present. The decrease in said metabolic activity was considered as a measure of antibacterial action of combinations of AMPs and antiseptics. Examples of typical curves obtained studying the metabolic activity of *E. coli* ML-35p exposed to the combinations of AMPs and antiseptics in sub-MIC concentrations equal to ¼ of their respective MICs are charted in Figure 2. Curves reaching the plateau correspond to the termination of the accumulation of a fluorescent product due to the complete depletion of the initial metabolic marker (in the case where the curve reaches the level demonstrated by the control of intact bacterium) or due to the complete suppression of metabolic processes in bacterial cells otherwise.

The effect of sub-MIC concentrations of antiseptics alone was less pronounced than that of antimicrobial peptides. The explanation may be that antiseptics’ “killing blow” on bacteria is delivered in a manner resembling an all-or-none law, with the fluctuation in susceptibility across the bacterial population being rather small, which is beneficial in terms of preventing bacteria from acquiring resistance. The concentration–effect relationship for AMPs was more gradual, although it reflected a faster and more pronounced action in sub-MIC doses. It was revealed that, in the presence of poviargolum, the development of the oppressive effect of both PG-1 and ChBac3.4 on the metabolic activity of bacteria was significantly accelerated, such that it became comparable to the action of ½ MIC of these peptides. A similar trend was also observed for PG-1 in combinations with dioxidyn and prontosan. Data on the action of etidronic acid were, unfortunately, unobtainable, due to the limitations of the selected redox reaction-based technique, since the compound is able to rapidly reduce resazurin to resorufin on its own regardless of the presence of actively metabolizing cells. Nevertheless, the observed dynamics clearly suggest that the addition of an antiseptic has a high chance of helping in narrowing the window of opportunity for the bacteria to slip away from the action of otherwise sublethal doses of AMPs.

### 3.6. Effects of the Individual Compounds and Their Combinations against Forming Monobacterial Biofilms

Biofilm formation is one of the defensive strategies that microorganisms frequently implement to resist the action of antimicrobial drugs and to protect themselves against effector cells and molecules of the immune system of the host [20,21,116]. Healthcare statistics suggest that between 60 and 80% of microbial infections, including nosocomial ones, are associated with the formation of biofilms [117,118]. On the wound surface, they postpone the healing process, and, on the surface of medical devices, they can inflict either inflammation that can ultimately result in rejection, if the device is an implant; damage to the device itself and reduction in its service life; or both [118,119,120]. The data available also indicate that polymicrobial biofilms provide a fertile environment for transmitting antimicrobial resistance genes across the bacterial species [120,121,122]. Preventing the formation of biofilms or destroying those already formed is an important task for successful antimicrobial therapy. AMPs and antiseptics are considered well-equipped and loaded with some additional “ammunition” while fighting against biofilms compared with commonly used antibiotics. In particular, the non-specific and multitargeted mode of action allows them to be equally effective against both actively metabolizing bacteria and the so-called persister cells, which are one of the safeguards ensuring the stability of biofilms and the chronic nature of the inflammatory process associated with it [21,123,124].

It was of interest to consider whether AMPs and antiseptics may enhance each other’s effects, not only toward planktonic bacteria but also toward biofilms. In this regard, we compared the ability of tested compounds to block biofilm formation while acting alone and in combinations. Among the clinical isolates that were previously exposed to AMP/antiseptic combinations in a form of suspension of free-swimming cells, a prominent ability to form monobacterial biofilms was demonstrated by *A. baumannii* 7226/16 and *P. aeruginosa* MDR 522/17. A weak biofilm could also be formed by *E. coli* ESBL 521/17, but only the first two bacteria were chosen for the test.

To quantitatively assess forming biofilms, matrix proteins and microbial cells adhered to the surface of the test wells were first stained with crystal violet, and, secondly, the bound dye was re-dissolved and colorimetrically evaluated.

Biofilm formation was found to be completely inhibited by individual substances in concentrations equal to or no more than two times higher than their MICs toward planktonic bacteria; rarely, mainly in the case of RFR-ChBac3.4(1–14), four-fold enhancement was required. Generally, a similar picture was revealed for the combinations as well: in cases where the concentrations of the components were sufficient to completely inhibit the growth of planktonic bacteria, biofilm formation was also not observed, which probably indicates the rapid development of the antimicrobial effect resulting in bacteria having no time to start forming the biofilm. Based on the results, we shifted our attention toward the effects of sub-microbicidal concentrations of the tested substances.

The minimal concentrations of AMPs and antiseptics causing a statistically significant inhibitory effect on biofilm formation (i.e., a statistically significant decrease compared with the control biofilm formed by the intact bacteria; Mann–Whitney U-test, *p* < 0.05) are shown in Table 3 and are later referenced as MEC_BF_ (for minimal effective concentration regarding biofilm formation). The exact value of such a minimal effect is admittedly small, within the range of a 10–20% decrease in the density of the emerging biofilm. Thus, the practical usefulness of such an effect, even if strengthened in combinations, is quite questionable; however, from the academic point of view, even slight changes can provide us with information on the mode of interaction between AMPs and antiseptics and allow us to make certain assumptions about the nature of such interactions.

Analyzing the combined action of AMPs and antiseptics (AS) in sub-microbicidal concentrations toward biofilms, we adopted the same approach based on the calculation of fractional indices as the one we used while testing antimicrobial activity on planktonic bacteria. For combinations, the series of concentrations from [MEC_BF_(AMP) + MEC_BF_(AS)], [½MEC_BF_(AMP) + ½MEC_BF_(AS)] and down to [1/32 × MEC_BF_(AMP) + 1/32 × MEC_BF_(AS)] was analyzed. In the case of poviargolum, the starting concentration was no more than ¼ MIC against both strains of bacteria, since higher concentrations, due to the synergistic interaction with AMPs, completely inhibited bacterial growth. The fractional indices, denoted here as fractional effective concentration indices (FECIs), were calculated as follows: FECI = [AMP]/MEC_BF_(AMP) + [AS]/MEC_BF_(AS), where [AMP] and [AS] are the concentration of AMP and antiseptic in a combination that has a statistically significant inhibitory effect on the formation of biofilms if compared with the intact control (Mann–Whitney U-test, *p* < 0.05). The minimal FECI > 4 was interpreted as antagonism, FECI ≤ 0.5 as synergy, 0.5 < FECI ≤ 1 as additivity, and 1 < FECI ≤ 4 as independent action.

Regarding the action of sub-microbicidal concentrations of tested compounds against forming biofilms (Table 4), a pronounced synergistic effect was established for combinations of all tested AMPs with prontosan and with cocamidopropyl betaine. The only exception was the effect of their combinations with RFR-ChBac3.4(1–14) toward *A. baumannii*, against which the AMP itself showed no antibiofilm effects in sub-MIC concentrations. In these cases, combined action was identified as additivity. It should be noted that, among the tested compounds, AMPs, amidopropyl betaine, and prontosan all showed the most noticeable tendency to suppress biofilm formation, even as individual agents. At the same time, prontosan also has an amidopropyl betaine in its composition as an auxiliary component; thus, it can be assumed that this component played a leading role in the detected antibiofilm synergy with AMPs. Based on the data on the mode of action of AMPs and antiseptics involved, our first guess on the probable mechanism is interference with primary adhesion of bacterial cells to contaminated surfaces.

In addition, synergy was also discovered for the combination of PG-1 with poviargolum, as well as of RFR-ChBac3.4(1–14) with dioxydin, toward the biofilms formed by *P. aeruginosa*. Considering that the last two compounds are believed to mainly affect intracellular targets, it can be assumed that their antibiofilm effect is also based primarily on blocking the expression of certain genes and proteins involved in biofilm formation.

Additivity of the antibiofilm effect, aside from the aforementioned combinations with RFR-ChBac3.4(1–14), was shown for combinations of PG-1 with dioxydin and ChBac3.4 and RFR-ChBac3.4(1–14) with poviargolum toward *P. aeruginosa* and also of ChBac3.4 with poviargolum and dioxydin toward *A. baumannii*. In the other cases, independent action was observed.

The experimental data confirmed AMPs’ and antiseptics’ prospects as antibiofilm agents and indicated that their antibiofilm properties could be further intensified in the presence of each other, at least concerning the biofilm formation stage. These results can be taken with a sufficient optimism regarding the potential of AMP/antiseptic implementation in coatings of medical devices, routine care products, or acute wound treatment. At the same time, literature data [22,125,126] suggest that eradicating well-established biofilms can require an order of magnitude higher concentrations of AMPs than the ones needed for the prophylactic against their formation or require prolonged exposure [127]. Thus, to properly assess the prospects of combined use of AMPs and antiseptics against such challenges, additional testing on preformed biofilms will be important.

### 3.7. Toxicity of the Individual Compounds and Their Combinations against Host Cells

Besides the prospects of overcoming bacterial resistance, combined use provides a potential opportunity to reduce unwanted side effects of the drugs (such as toxicity) in cases where one will be able to find a combination in which the components selectively enhance only the desired activity of each other and do not enhance toxic effects against the host’s body. Toxicity of the tested compounds and their combinations was evaluated against human red blood cells and human dermal fibroblasts. The hemolytic test was considered, to a certain extent, as an indicator of a non-selective promotion of the membranolytic properties that are regarded as the main cause of the toxicity of AMPs toward eukaryotic cells in cases where it is identified. The choice of fibroblasts as target cells was dictated by the fact that topical treatment is the most probable niche for the combined application of AMPs and antiseptics.

The scheme of evaluating combined cytotoxic action was similar to the one used to analyze the joint effect of AMPs and antiseptics against biofilms: minimal effective cytotoxic/hemolytic concentrations of individual substances (MEC) were determined (i.e., the minimal concentrations that cause an effect statistically different from the reference value provided by the control of intact cells), and then the fractional indices were calculated, assessing the presence or absence of statistically significant toxicity for their combinations in individually sub-cytotoxic doses. While analyzing toxicity toward eukaryotic cells, components were mixed in concentrations of ½ MEC (AMP) + ½ MEC (AS) (that can provide the fractional index = 1, denoting additivity, if found effective) and ¼ MEC (AMP) + ¼ MEC (AS) (that can provide the fractional index = 0.5, denoting synergy, if found effective).

#### 3.7.1. Hemolytic Activity

Hemolytic activity was evaluated using a standard hemolytic test: hemoglobin escaping red blood cells damaged during the incubation with test substances or their combinations was colorimetrically quantified, and its amount was interpreted as a percentage of affected cells.

Minimal concentrations inflicting hemolytic effect (MEC_H_), detectable once statistically compared with the intact control (Mann–Whitney U-test, *p* < 0.05), are given in Table 3. The data obviously show that for only half of the tested substances did their individual minimal hemolytic concentrations exceed any of their MICs toward bacteria. This was true for proline-rich AMP ChBac3.4 and its shortened modification RFR-ChBac3.4(1–14) (MEC_H_ was higher than 40–50 µM) and for dioxydin and poviargolum among antiseptics (MEC_H_ was higher than 0.5 mg/mL). This fact emphasizes the importance of reducing toxic effects of AMPs and antiseptics.

The absence “–” or presence “+” of a statistically significant difference compared with the intact control once exposed to the action of combinations of AMPs and antiseptics in concentrations equal to ½ or ¼ of their individual MEC_H_ was evaluated in a series of three independent experiments. The data are summarized in Table 5; the resulting assessment for the series is given in square brackets. The fractional indices (here FECI, calculated similarly to the ones describing combined effect on biofilm formation) that correspond to synergistic interaction are given in bold. Synergy of hemolytic action is revealed only for the combination of the membranolytic peptide PG-1 with prontosan. Notably, both these substances show a pronounced individual hemolytic effect.

The hemolytic activity of prontosan (that includes undecylenamidopropyl betaine) turned out to be very similar to the hemolytic activity of the amidopropyl betaine surfactant that we tested alone. This suggests that prontosan’s hemolytic properties should be provided mainly by this component. On the other hand, in combinations with amidopropyl betaine, protegrin and other AMPs showed only additivity. Thus, additional influence from polyhexanide cannot be excluded. The remaining combinations of AMPs and antiseptics mainly provided an additive effect as well. No increase in hemolytic capabilities upon combined use was found only for combinations of proline-rich peptide ChBac3.4 and its shortened analog RFR-ChBac3.4(1–14) with dioxydin and etidronic acid.

#### 3.7.2. Cytotoxic Action toward Human Dermal Fibroblasts

Trying to take a brief glance at the possible source of the detected synergy cases between AMPs and antiseptics, we started with assessing the changes in membranolytic capabilities of the compounds used in combinations, as the disruption of bacterial membranes is the fastest killing scenario performed by most AMPs capable of it.

Cytotoxic effects of individual substances and their combinations were also evaluated against cultured normal human dermal fibroblasts. We used MTT tests and/or light microscopy for the direct cell count. The latter was used primarily to assess the toxic action of etidronic acid (or of the combinations with it), since during the resazurin assay we uncovered that this agent was capable of unconditionally converting a redox-dependent dye designed to otherwise reflect the activity of the respiratory chain. MTT works on the same principle and, hence, may be compromised.

Minimal effective concentrations of individual substances (MEC_F_) are included in Table 3, and the data considering combined action are summarized in Table 6 in the same manner as the one used for the hemolytic test. The overall picture is also similar. PG-1, which has prominent hemolytic activity, began to exert a statistically significant toxic effect toward human dermal fibroblasts in concentrations above 5 µM, and at the concentration of 20 μM its magnitude became higher than 50%.

The antiseptic agent prontosan, which also possesses notable hemolytic properties, demonstrated no toxicity up to the concentration of 5 μg/mL, but with a further increase in concentration its cytotoxic effect also intensified rapidly and exceeded 50% at 30 μg/mL. At the same time, in contrast to the hemolytic test results, the joint effect of PG-1 and prontosan in sub-cytotoxic concentrations was only additive. The impact of etidronic acid on fibroblasts was even more pronounced than on erythrocytes, probably due to the pH effect.

Given its MEC_F_ (minimal effective concentration against fibroblasts) of less than 2.5 mg/mL (where the effect was already higher than 50%), which is also below any of its MICs against bacteria, it was excluded from the further analysis of the toxicity in combinations with AMPs. The individual toxic effects of dioxidyn, as well as of the proline-rich AMPs, were low: MEC_F_ for dioxydin was 0.5 mg/mL; for ChBac3.4, it equaled 40 µM, and for RFR-ChBac3.4(1–14) it was even higher. However, the cytotoxic action of poviargolum against fibroblasts, contrary to its small hemolytic activity, manifested quite prominently. Light microscopy performed after 30 min of incubation with poviargolum showed that fibroblasts disadhered from the surface of the test wells.

Although no cases of synergy of cytotoxic action against fibroblasts was identified, additivity nevertheless prevailed over the complete absence of any amplification of the individual effects. Independent action was found for combinations of all AMPs with dioxydin and of RFR-ChBac3.4(1–14) with betaine.

Admittedly, toxicity tests were performed in a serum-free environment over quite long periods of time, which is not exactly ideal for cultured cells’ well-being. Thus, it is rather probable that in vivo toxic concentrations will be higher, whereas cytotoxicity will be, respectively, lower.

## 4. Discussion

The powerful biocidal effect of antiseptics is a significant bonus in combating microorganisms, overcoming their resistance mechanisms, including biofilms. In many cases their action has a more universal physicochemical basis rather than a quite refined and subtle biochemical pathway typical of classic antibiotics. While it contributes to their broad spectrum of activity and greatly interferes with bacteria’s efforts to design suitable resistance strategies—so much so that for many antiseptics no resistance cases are reported despite decades of use—it is also a key factor defining their toxicity issues that place significant restrictions on the area of their medical use. The fact that they are virtually unsuitable for general administration and often cannot be delivered to the site of infection within the body is, probably, the main reason why they mainly stay in the shadow of antibiotics in the public and even scientific eye.

While the idea of a combined use of AMPs with other antimicrobials as a step toward their integration into practice is not new, it is also more focused on coupling them with antibiotics, whereas the data on their interaction with antiseptics, aside from nanoparticles, seem to be few. Though this state of affairs is understandable, in a narrower niche of topical application, antiseptics may prove to be valuable assets and deserve their share of trials.

Actually, the usefulness of the information on the interaction between AMPs and antiseptics is not limited to the prospects of enhancing their potential against bacteria and their resistance arsenal, with a boon of dose and toxicity reduction for creating new drug combinations or hybrid coatings for medical devices or dressings.

First, at the late stages of clinical trials where actual patient welfare may be at risk (e.g., the treatment of gangrene in diabetic patients), the most ethical choice of trial design is to compare the standard therapy with the combination of the standard therapy with the new one [128]. Antiseptics are common participants in the standard topical treatment [34,129,130]. Thus, synergy or antagonism of a new AMP-based drug with them may bias trial outcomes and should be taken into consideration, preferably at the stage of trial design.

Second, in the case of AMP-based coatings on medical products, AMPs’ nature already limits the available disinfection/sterilization methods to the soft chemical ones. Diminishing the activity of AMPs upon contact with components of a disinfecting solution (antiseptics or surfactants), especially if permanent, will be a truly unfavorable result. For example, if AMPs are used as a coating for contact lenses [33], this situation can occur during their cleaning and maintenance.

Additionally, some autoimmune and/or skin diseases, such as psoriasis, atopic dermatitis, rosacea, acne vulgaris, and burn and chronic wounds, are reported to be accompanied by an alteration to the level of AMP production [131,132,133]. Cases of both over- and under-expression of AMPs have been reported. Though it remains unclear whether the shift in AMP balance contributes to the pathogenesis or is merely a collateral symptom or a protection mechanism in each particular case, the possibility to enhance or decrease the activity of AMPs by other compounds may be useful in designing the complex treatment.

Between AMPs and antiseptics, even within the limited set that we tested in our study, both synergy and antagonism were indeed identified. This is quite different compared with the statistics on AMPs’ interactions with antibiotics, where no distinctly antagonistic cases were found, to our knowledge.

In general, synergy is explained by the scenario where one component helps another to reach its site of action, including interference with the target’s attempts at resistance, or the scenario where both components inhibit or activate alternative routes contributing to the desired effect [45,51,134]. Antagonism, in turn, is regarded as a situation where one component prevents the other from interacting with its target—by extension, an outcome where two components form an inactive product upon direct interaction [135].

Defined antagonistic cases seem to fit well into the described pattern. Sodium hypochlorite can directly damage peptide and protein molecules [78], and both hypochlorite and sarkosyl anions may interfere with the electrostatic attraction of cationic AMPs to the anionic groups of the compounds layering the surface of bacterial cells.

Synergy between AMPs and antiseptics is, however, more difficult to interpret, due to a multitarget mode of action of both of these groups of compounds. We may deduce some of the underlying mechanics. For example, it is known that bisphosphonates are highly hydrophilic compounds; even their absorption in the intestine occurs by paracellular transport [94,136]. Thus, etidronic acid should obviously benefit from the enhanced permeability of bacterial membranes inflicted by AMPs. This also applies for other antiseptics not showing membranolytic effects by themselves, such as polyhexanide (prontosan) or dioxydin.

Moreover, extensive membrane damage should drastically affect the effectiveness of a number of defensive mechanisms of bacteria, including efflux pumps, which are a quite universal route of eliminating various harmful compounds from bacterial cells [137]. For example, the role of efflux systems is identified in bacterial resistance to silver compounds [87,138,139]. Antiseptics, for their part, also can disarrange their fair share of the resistance machinery: e.g., inactivate certain enzymes or block the inducible mechanisms on the gene transcription stage.

In addition, bisphosphonates are able to bind Ca^2+^ ions [94]. Bivalent cations, such as Ca^2+^ and Mg^2+^, are known to interact with negatively charged phosphate groups of the bacterial outer membrane and to create an electrostatic barrier preventing hydrophilic compounds from penetrating within the cell [16]. AMPs are believed to compete with such cations for the electrostatic binging on the bacterial surface. Enhanced concentrations of bivalent cations have been shown to diminish AMP activity [140,141]. Thus, the decrease in concentration of free Ca^2+^ should have the opposite effect. In this aspect, the observed synergy with etidronic acid can be compared to the synergistic effect reported for AMPs with EDTA or other chelators that is attributed to the ability of the latter to bind divalent metals (Ca^2+^, Mg^2+^, Zn^2+^, etc.) from the growth medium [142,143].

Amidopropyl betaine surfactant seems to more readily damage eukaryotic membranes than bacterial ones if alone, comparing its MICs with its hemolytic concentration. We may presume that AMPs can act as a “fuse” of some sort, causing an initial disturbance in the bacterial membrane that makes it more susceptible to betaine.

In retrospect, many of the observed effects are in a good correlation with general conceptions regarding the influence of various basic environmental factors, such as pH, cation or anion presence, etc., on the activity of AMPs [142,144], and with the ability of antiseptics to denature proteins and, hence, incapacitate AMPs directly. This suggests that our basic notions on AMPs’ and antiseptics’ modes of action have a sufficient level of predictive power for the preliminary assessment of new combinations. A number of the identified effects manifest rather consistently among different AMPs; some synergy cases also translate well from planktonic bacteria to forming biofilms. All of this casts a positive light on the prospects of the combined use of AMPs and antiseptics.

At the same time, analyzing the general picture, we should note that our initial idea of verifying which mode of action of the AMPs was more favorable for synergy with certain antiseptics by cross comparing PG-1 with ChBac3.4, which are capable of membranolytic action, and ChBac3.4 with RFR-ChBac3.4(1–14), which can alter intracellular processes, did not pay off. There is little correlation in the synergy manifestation in both pairs. The situation is quite different from what was found previously for ChBac3.4 and RFR-ChBac3.4(1–14) when comparing their combined action with a number of antibiotics; the synergy profiles were almost identical in that case [47]. On top of that, more synergy cases were found for a non-membranolytic RFR-ChBac3.4(1–14): 66,7% (16 out of 24) vs. 50% (15 out of 30) for ChBac3.4 and 40% (12 out of 30) for PG-1. This leads us to the hypothesis that the observed synergy may rely more on the killing kinetics rather than on the particular mode of action. In other words, the quicker the compound delivers its bactericidal effect, the shorter is the time lapse for the other compound to contribute some sort of assistance in the process. On the one hand, it is not ideal for the good predictability of synergy cases of new AMPs with the same antiseptics based on the data obtained for other AMPs, even close derivatives. On the other, if true, it places non-membranolytic AMPs in a favorable position. While they generally have low toxicity, there is always a concern that, due to their more target-specific action, it will be easier for the bacteria to acquire resistance to them. Thus, such AMPs may gain the most from the combined action with other antimicrobials, including antiseptics, in terms of safe use. This can allow both lowering the possibility of resistance emergence and having a better chance to reduce overall toxicity of the combination.

However, our in vitro observations of the combined toxicity of AMPs and antiseptics against eukaryotic cells indicate that the possible reduction will be rather small. Most combinations demonstrate at least additivity of hemolytic action and of cytotoxic effect toward human fibroblasts in culture. On the other hand, despite low toxic concentrations of some antiseptics and surfactants that we established in vitro, the fact that they are considered safe for topical use in cosmetic and healthcare products inspires more optimistic expectations for the effect on full skin.

Nevertheless, our data suggest that AMP/antiseptic interactions provide interesting possibilities for designing new effective therapeutics and medical materials. In the era of bacterial resistance crisis, no compounds with potentially low resistance rates should be overlooked, and the question of combined use of AMPs and antiseptics certainly deserves further and broader investigation.

## 5. Conclusions

Summarizing the practical tips from our AMP/antiseptic and surfactant interaction analysis, we can say that, in designing topical antimicrobial compositions or hybrid antibiofilm coatings, AMPs with membranolytic capacities show a greater promise against biofilm formation but also should be verified for toxicity enhancement, especially if the end product is meant to be applied for prolonged periods of time. On the other hand, combining antiseptics with synergistic non-toxic AMPs aiming at intracellular targets in bacteria may provide the best toxicity reduction and lessen the concerns regarding the probability of resistance emergence toward the latter.

Finally, we should stress that, because not only synergy but also antagonism is a possibility between the AMPs and antiseptics or surfactants, the situations where these compounds come into contact need to be carefully considered: from the designing and storage of AMP-including compositions, where it is an obvious part of the development, to the disinfection and maintenance of AMP-coated medical devices or AMP-based product application alongside other treatment procedures that may include the use of antiseptics, where such interaction is easier to overlook. Highly anionic or denaturizing compounds are the most certain candidates to be avoided.

## Figures and Tables

**Figure 1 pharmaceutics-15-00291-f001:**
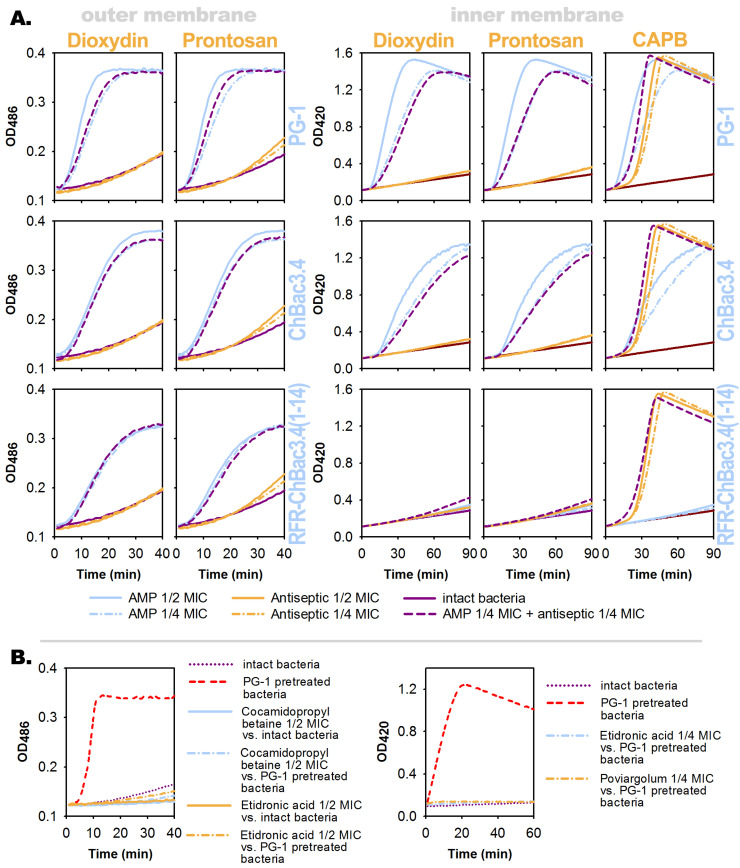
(**A**) Effects of combinations of AMPs with antiseptics and surfactants on membrane permeability of *E. coli* ML-35p. Individual action of ¼ and ½ of minimal inhibitory concentrations (MICs) of compounds is compared to that of their combinations containing ¼ MIC of AMP + ¼ MIC of antibiotic/surfactant. Curves illustrate nitrocefin degradation by the bacterial periplasmic β-lactamase (for the outer membrane permeability assessment) or o-nitrophenyl-β-D-galactoside (ONPG) degradation by the bacterial cytoplasmic β-galactosidase (for inner membrane permeability assessment) which occur when the membranes are damaged by test substances. The steeper slope of the curve and the shorter time to reach the plateau level correspond to a greater extent of damage. CAPB—cocamidopropyl betaine. (**B**) Interference of some of the tested compounds with marker molecules’ transformation by bacterial enzymes.

**Figure 2 pharmaceutics-15-00291-f002:**
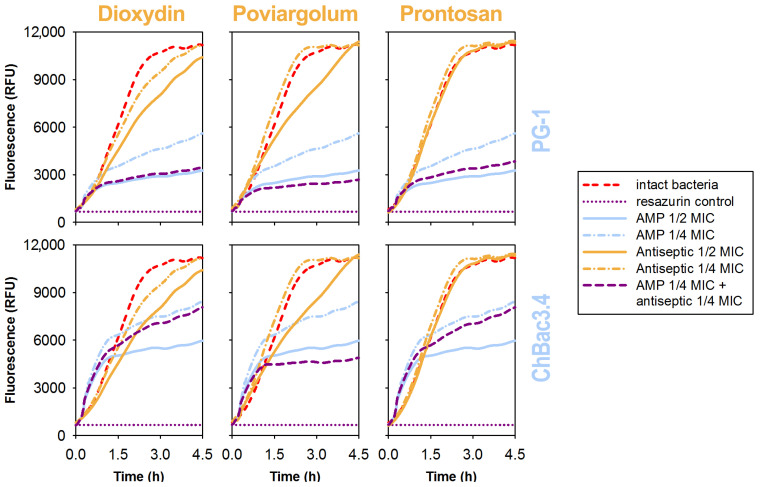
Dynamics of *E. coli* ML-35p metabolic activity inhibition by the combinations of AMPs with antiseptics and surfactants. Individual action of ¼ and ½ of minimal inhibitory concentrations (MICs) of compounds is compared to that of their combinations containing ¼ MIC of AMP + ¼ MIC of antibiotic/surfactant. An increase in fluorescence intensity indicates actively metabolizing bacteria in which redox-sensitive marker resazurin is continuously reduced into fluorescent resorufin. The quicker the plateau is reached and the lower its level is compared with the control of intact bacteria, the stronger and faster is the antimicrobial effect that it suggests.

**Table 1 pharmaceutics-15-00291-t001:** Antimicrobial activity of AMPs, antiseptics, and surfactants alone against drug-resistant clinical isolates and *E. coli* ML-35p laboratory strain.

Substances	MIC ^a^ against Drug-Resistant Bacteria					
	Gram−					Gram+
	*E. coli * ML-35p	*E. coli * ESBL 521/17	*A. baumannii* 7226/16	*P. aeruginosa* MDR 522/17	*K. pneumoniae* ESBL 344/17	*S. aureus * 1399/17
AMPs (µM) ^b^:						
PG-1	0.8	0.4	6.2	6.2	3.1	0.4
ChBac3.4	1.6	3.1	6.2	6.2	3.1	3.1
RFR-ChBac3.4 (1–14)	3.1	6.2	12.5	16	6.2	6.2
Antiseptics (µg/mL):						
Dioxydin	15.6	15.6	31.2	125	15.6	500
Poviargolum	39.1	39.1	39.1	39.1	39.1	156.2
Prontosan *	0.4	0.4	31.2	3.1	1.6	0.8
Etidronic acid	50,000	50,000	12,500	25,000	50,000	50,000
Sodium hypochlorite	5000	5000	2500	5000	5000	10,000
Surfactants (µg/mL):						
Cocamidopropyl betaine	>50,000	>50,000	78.1	>50,000	156.2	>50,000
Sodium lauroyl sarcosinate	6250	12,500	25,000	50,000	12,500	>50,000

^a^ Minimal inhibitory concentration (MIC) values are medians of 3–4 independent experiments made in triplicates. If MIC was not found within the tested concentration range, it was presumed to be twice the maximal tested concentration in the experiments regarding combined effect. ^b^ AMP concentrations are given in µM for easier comparison with the results for other AMPs without taking into account the difference in their molecular weight (MW). To transform them into µg/mL, one should increase the µM values by a factor of MW∙10^−3^, which is 2.2 for PG-1, 3.4 for ChBac3.4, and 2.3 for RFR-ChBac3.4(1–14). * Specified prontosan concentrations are concentrations of its main antimicrobial component—polyhexanide. It also contains undecylenamidopropyl betaine in equal concentration.

**Table 2 pharmaceutics-15-00291-t002:** Antimicrobial activity of combinations of AMPs with antiseptics and surfactants against drug-resistant clinical isolates and *E. coli* ML-35p laboratory strain.

Minimal FICIs ^a^ of Surfactants or Antiseptics (AS) Combinations with AMPs against Drug-Resistant Bacteria											
*E. coli* ML-35p (Gram–)						*E. coli* ESBL 521/17 (Gram–)					
**AMP\AS**	DXD	PVG	PTS	CAPB	ETA	**AMP\AS**	DXD	PVG	PTS	CAPB	ETA
PG-1	*1.12*	**0.5**	**0.5**	**0.31**	**0.5**	PG-1	*1.12*	0.56	0.75	**0.25**	0.56
ChBac3.4	0.62	0.75	0.75	**0.38**	**0.5**	ChBac3.4	0.75	**0.38**	1	**0.38**	**0.25**
RFR-ChBac3.4 (1–14)	0.75	0.62	0.75	**0.38**	-	RFR-ChBac3.4 (1–14)	**0.5**	**0.5**	0.56	**0.38**	-

*A. baumannii* 7226/16 (Gram–)						*P. aeruginosa* MDR 522/17 (Gram–)					
**AMP\AS**	DXD	PVG	PTS	CAPB	ETA	**AMP\AS**	DXD	PVG	PTS	CAPB	ETA
PG-1	0.75	0.62	0.62	**0.5**	1	PG-1	1	**0.5**	1	0.56	*1.12*
ChBac3.4	**0.5**	0.75	0.75	**0.38**	**0.5**	ChBac3.4	1	0.62	0.75	0.56	0.75
RFR-ChBac3.4 (1–14)	*1.12*	**0.38**	**0.38**	**0.5**	-	RFR-ChBac3.4 (1–14)	**0.31**	**0.25**	**0.31**	**0.25**	-

*K. pneumoniae* ESBL 344/17 (Gram–)						*S. aureus* 1399/17 (Gram+)					
**AMP\AS**	DXD	PVG	PTS	CAPB	ETA	**AMP\AS**	DXD	PVG	PTS	CAPB	ETA
PG-1	0.62	**0.5**	0.75	**0.38**	*1.12*	PG-1	0.62	**0.5**	1	**0.31**	**0.5**
ChBac3.4	**0.5**	**0.5**	0.62	**0.38**	1	ChBac3.4	0.75	**0.5**	**0.38**	**0.5**	**0.38**
RFR-ChBac3.4 (1–14)	0.62	0.62	**0.5**	**0.38**	-	RFR-ChBac3.4 (1–14)	**0.5**	**0.25**	0.75	**0.12**	-


DXD—dioxydin; PVG—poviargolum; PTS—prontosan; CAPB—cocamidopropyl betaine; ETA—etidronic acid. ^a^ Fractional inhibitory concentration index (FICI) values are medians of 3 independent experiments. FICI > 2 specifies antagonistic interaction; 1 < FICI ≤ 2 indicates independent action; 0.5 < FICI ≤ 1 is designated as additivity; FICI ≤ 0.5 is regarded as synergy. Synergy cases are set off in bold type; cases of independence are given in italic. Table cells are colored according to the heat maps generated using the Morpheus online tool (available at https://software.broadinstitute.org/morpheus/, accessed on 25 December 2022). Color changes from green (synergy) through yellow (additivity) to orange-red (independence and antagonism).

**Table 3 pharmaceutics-15-00291-t003:** Individual effects of AMPs, antiseptics, and surfactants against forming biofilms and eukaryotic cells (human erythrocytes and human dermal fibroblasts).

Substances	Effects of Sub-MIC Concentrations on Biofilm Formation		Toxic Effects against Human Cells	
	MEC_BF_ ^a^ Partially Inhibiting Bacterial Biofilms		MEC_H_ ^a^ of Hemolysis of Human Erythrocytes	MEC_F_ ^a^ of Cytotoxic Action toward Normal Human Dermal Fibroblasts
	*A. baumannii* 7226/16	*P. aeruginosa* MDR 522/17		
*AMPs:*	MIC ratio		µM ^b^	
PG-1	1/32	1/64	**0.5**	**5**
ChBac3.4	1/16	1/64	>40 {80}	40
RFR-ChBac3.4(1–14)	1	1/128	>50 {100}	>40 {80}
Antiseptics & surfactants:	MIC ratio		µg/mL	
Dioxydin	1/16	1/16	500	500
Poviargolum	1	1/4	500	**12.5**
Prontosan	1/16	1/256	**1.6 ***	**3.1 ***
Etidronic acid	1/128	1/32	**5000**	**<2500**
Cocamidopropyl betaine	1/32	1/512	**1**	**12.5**

^a^ Minimal effective concentrations (MECs) reducing the thickness of the forming biofilms (BF) or inducing hemolysis (H) or cytotoxic action against fibroblasts (F) are medians of 3 independent experiments. MECs are minimal concentrations where the statistically significant difference from the untreated (intact) control is found with the Mann–Whitney U-test (*p* < 0.05; n_1_ = 4, n_2_ = 8 for antibiofilm action; n_1_, n_2_ = 3 for hemolysis; n_1_ = 3, n_2_ = 6–8 for cytotoxic action). If MEC was higher than the maximal concentration tested, it was presumed to be twice this concentration in the experiments regarding combined effect and is given in curly brackets. If MEC_H_ or MEC_F_ is lower than some of the MICs established against bacteria, it is given in bold. ^b^ AMP concentrations are given in µM for easier comparison with the results for other AMPs without taking into account the difference in their molecular weight (MW). To transform them into µg/mL, one should increase the µM values by a factor of MW∙10^−3^, which is 2.2 for PG-1, 3.4 for ChBac3.4, and 2.3 for RFR-ChBac3.4(1–14). * Specified prontosan concentrations are concentrations of its main antimicrobial component—polyhexanide. It also contains undecylenamidopropyl betaine in equal concentration.

**Table 4 pharmaceutics-15-00291-t004:** Inhibitory effect of combinations of AMPs with antiseptics and surfactants on biofilm formation.

Minimal FECIs ^a^ of Surfactants or Antiseptics (AS) Combinations with AMPs Partially Inhibiting Bacterial Biofilms’ Formation											
*A. baumannii* 7226/16 (Gram–)						*P. aeruginosa* MDR 522/17 (Gram–)					
**AMP\AS**	DXD	PVG	PTS	CAPB	ETA	**AMP\AS**	DXD	PVG	PTS	CAPB	ETA
PG-1	*1.12*	*1.12*	**0.25**	**0.5**	*1.12*	PG-1	1	**0.25**	**0.12**	**0.12**	*2*
ChBac3.4	1	1	**0.5**	**0.5**	*1.12*	ChBac3.4	*1.12*	1	**0.12**	**0.5**	*2*
RFR-ChBac3.4 (1–14)	*1.12*	*1.12*	1	1	-	RFR-ChBac3.4 (1–14)	**0.5**	1	**0.12**	**0.5**	-


DXD—dioxydin; PVG—poviargolum; PTS—prontosan; CAPB—cocamidopropyl betaine; ETA—etidronic acid. ^a^ Fractional effective concentration index (FECI) values are medians of 3 independent experiments. FECI > 4 specifies antagonistic interaction; 1 < FECI ≤ 4 indicates independent action; 0.5 < FECI ≤ 1 is designated as additivity; and FICI ≤ 0.5 is regarded as synergy. Synergy cases are set off in bold type; cases of independence are given in italic. Table cells are colored according to the heat maps generated using the Morpheus online tool (available at https://software.broadinstitute.org/morpheus/, accessed on 25 December 2022). Color changes from green (synergy) through yellow-orange (additivity) to orange-red (independence).

**Table 5 pharmaceutics-15-00291-t005:** Hemolytic effect of combinations of AMPs with antiseptics and surfactants toward human erythrocytes.

AMP (A)	Hemolytic Action of (½ MEC A + ½ MEC B) and (¼ MEC A + ¼ MEC B) Combinations toward Human Erythrocytes and Corresponding FECIs ^a^														
	Antiseptic (B)														
	Dioxydin			Poviargolum			Prontosan			Cocamidopropyl Betaine			Etidronic Acid		
	**½** **A & ½** **B**	**¼** **A & ¼** **B**	**FECI**	**½** **A & ½** **B**	**¼** **A & ¼** **B**	**FECI**	**½** **A & ½** **B**	**¼** **A & ¼** **B**	**FECI**	**½** **A & ½** **B**	**¼** **A & ¼** **B**	**FECI**	**½** **A & ½** **B**	**¼** **A & ¼** **B**	**FECI**
PG-1	+ + − [+]	− − − [−]	1.0	+ + − [+]	− − − [−]	1.0	+ + + [+]	+ + + [+]	**0.5**	+ + + [+]	+ − − [−]	1.0	+ + + [+]	− − − [−]	1.0
ChBac3.4	− − − [−]	− − − [−]	>1.0	+ + + [+]	− − − [−]	1.0	+ + − [+]	− − − [−]	1.0	+ + − [+]	− − − [−]	1.0	− − − [−]	− − − [−]	>1.0
RFR-ChBac3.4 (1–14)	+ − − [−]	+ − − [−]	>1.0	− − − [−]	− − − [−]	>1.0	+ + + [+]	− − − [−]	1.0	+ + + [+]	+ − − [−]	1.0			

+|− presence or absence of statistically significant difference compared with the control of intact cells (Mann–Whitney U-test, *p* < 0.05) in each of the 3 independent experiments; the overall result is given in square brackets. ^a^ Assessment of the minimal fractional effective concentration index (FECI) based on the results for (½ MEC A + ½ MEC B) and (¼ MEC A + ¼ MEC B) combinations of substances A and B. FECI > 1.0 indicates independent action or antagonism; 0.5 < FECI ≤ 1 specifies additivity; FECI ≤ 0.5 denotes synergy. Synergy cases are given in bold type. Table cells are colored according to the heat maps generated using the Morpheus online tool (available at https://software.broadinstitute.org/morpheus/, accessed on 25 December 2022). Color changes from red-orange (synergy) through yellow (additivity) to green (independence or antagonism). Color map is reversed compared to the antimicrobial activity testing, as the desired result here is the absence of synergy. MEC—minimal effective concentration.

**Table 6 pharmaceutics-15-00291-t006:** Toxicity of combinations of AMPs with antiseptics and surfactants toward human dermal fibroblasts.

AMP (A)	Toxicity of (½ MEC A + ½ MEC B) and (¼ MEC A + ¼ MEC B) Combinations toward Human Dermal Fibroblasts and Corresponding FECIs ^a^											
	Antiseptic (B)											
	Dioxydin			Poviargolum			Prontosan			Cocamidopropyl Betaine		
	½ A & ½ B	¼ A & ¼ B	FECI	½ A & ½ B	¼ A & ¼ B	FECI	½ A & ½ B	¼ A & ¼ B	FECI	½ A & ½ B	¼ A & ¼ B	FECI
PG-1	− − − [−]	− − − [−]	>1.0	+ + + [+]	+ − − [−]	1.0	+ + − [−]	− − − [−]	1.0	+ + + [+]	+ − − [−]	1.0
ChBac3.4	− − − [−]	− − − [−]	>1.0	+ + + [+]	+ − − [−]	1.0	+ + + [+]	+ − − [−]	1.0	+ + + [+]	− − − [−]	1.0
RFR-ChBac3.4 (1–14)	+ − − [−]	+ − − [−]	>1.0	+ + + [+]	+ − − [−]	1.0	+ + + [+]	+ − − [−]	1.0	+ − − [−]	+ − − [−]	>1.0

+|− presence or absence of a statistically significant difference compared with the control of intact cells (Mann–Whitney U-test, *p* < 0.05) in each of the 3 independent experiments; the overall result is given in square brackets. ^a^ Assessment of the minimal fractional effective concentration index (FECI) based on the results for (½ MEC A + ½ MEC B) and (¼ MEC A + ¼ MEC B) combinations of substances A and B. FECI > 1.0 indicates independent action or antagonism; 0.5 < FECI ≤ 1 specifies additivity; FECI ≤ 0.5 denotes synergy. Table cells are colored according to the heat maps generated using the Morpheus online tool (available at https://software.broadinstitute.org/morpheus/, accessed on 25 December 2022). Color changes from red-orange (synergy) through yellow (additivity) to green (independence or antagonism). Color map is reversed compared to the antimicrobial activity testing, as the desired result here is the absence of synergy. MEC—minimal effective concentration.

## Data Availability

The data presented in this study are available on request from the corresponding author. Authors confirm that all relevant data supporting the findings of this study are included in the article.

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
