# Peer review of "Combined Use of Antimicrobial Peptides with Antiseptics against Multidrug-Resistant Bacteria: Pros and Cons"

_pharmaceutics, 2023, doi:10.3390/pharmaceutics15010291_

Round 1

Reviewer 1 Report

Dear Authors,

Presented manuscript is a well-written study on combinational approach between antimicrobial peptides (AMPs) and antiseptics. The style like to whole follow-up of the manuscript is very elegant so it was realy easy to follow.

I have no doubts that the data will be usefull for scientific community.

Therefore thera are some minor points, that in my opinion could be revised. They are listed below:

1. In introduction Line 90. There is a statement that .."[...] They are also regarded as promising antibiofilm agents [39,40].", but the references are connected with antiiofilm activity of antiseptics. In my opinion it woudl be good to enrich this part in some information about anibiofilm capabilities of AMPs. Please see:

1)Pletzer, D., Hancock, R.E.W Antibiofilm peptides: Potential as broadspectrum agents (2016) Journal of Bacteriology, . DOI: 10.1128/JB.00017-16

2)De La Fuente-Núñez, C., Cardoso, M.H., De Souza Cândido, E., Franco, O.L., Hancock, R.E.W.Synthetic antibiofilm peptides(2016) Biochimica et Biophysica Acta - Biomembranes, . DOI: 10.1016/j.bbamem.2015.12.015

3)Crabbé, A., Liu, Y., Matthijs, N., Rigole, P., De La Fuente-Nùñez, C., Davis, R., Ledesma, M.A., Sarker, S., Van Houdt, R., Hancock, R.E.W., Coenye, T., Nickerson, C.A.Antimicrobial efficacy against Pseudomonas aeruginosa biofilm formation in a three-dimensional lung epithelial model and the influence of fetal bovine serum.(2017) Scientific Reports, . DOI: 10.1038/srep43321

4) Jaśkiewicz, M., Neubauer, D., Kazor, K., Bartoszewska, S., Kamysz, W.Antimicrobial Activity of Selected Antimicrobial Peptides Against Planktonic Culture and Biofilm of Acinetobacter baumannii(2019) Probiotics and Antimicrobial Proteins, . DOI: 10.1007/s12602-018-9444-5

5)Flemming K, Klingenberg C, Cavanagh JP, Sletteng M, Stensen W, Svendsen JS, Flaegstad T. High in vitro antimicrobial activity of synthetic antimicrobial peptidomimetics against staphylococcal biofilms. J Antimicrob Chemother. 2009 Jan;63(1):136-45. doi: 10.1093/jac/dkn464. Epub 2008 Nov 14. PMID: 19010828.

6)de Freitas LM, Lorenzón EN, Cilli EM, de Oliveira KT, Fontana CR, Mang TS. Photodynamic and peptide-based strategy to inhibit Gram-positive bacterial biofilm formation. Biofouling. 2019 Aug;35(7):742-757. doi: 10.1080/08927014.2019.1655548. Epub 2019 Sep 5. PMID: 31550929.

2. Line 157. Methods for examing antibacterial effects.

There is no need to add information on percentage (2,1%) of MHB. Just MHB, as it well descrbed in literature and in broth-microdilution protocols.

3.Why you did not consider evaluation of antibiofilm activity by determination of minimal biofilm eradication concentration (MBEC) values alone and/or in combination (since you have used resazruin and or crystal violet in your assays, please see :

1)Pettit RK, Weber CA, Kean MJ, Hoffmann H, Pettit GR, Tan R, Franks KS, Horton ML. Microplate Alamar blue assay for Staphylococcus epidermidis biofilm susceptibility testing. Antimicrob Agents Chemother. 2005 Jul;49(7):2612-7. doi: 10.1128/AAC.49.7.2612-2617.2005. PMID: 15980327; PMCID: PMC1168683.

2)Peeters E, Nelis HJ, Coenye T. Comparison of multiple methods for quantification of microbial biofilms grown in microtiter plates. J Microbiol Methods. 2008 Feb;72(2):157-65. doi: 10.1016/j.mimet.2007.11.010. Epub 2007 Nov 21. PMID: 18155789.

3)Jaśkiewicz, M.; Janczura, A.; Nowicka, J.; Kamysz, W. Methods Used for the Eradication of Staphylococcal Biofilms. Antibiotics 2019, 8, 174. https://doi.org/10.3390/antibiotics8040174).

In my opinion not alwas sub-inhibitory concentrations will give insights into antibiofilm potential. It must be taken into account that biofilm is a multilayered formation and peptides/antiseptics do not penetrate and disrupt biofilm in the linear manner. This should be discussed. 

3. In my opinion presentation of the results included  in Tables: 2,4,5, and 6 would be more elegant, when presented on heat maps, for instance please use: https://software.broadinstitute.org/morpheus/

I have no other considerations regarding the manuscript. 

Best regards

Author Response

Dear Reviewer,

We are very grateful for this positive overview and very valuable suggestions!

We made some changes in the manuscript accorging to the Reviewer's comments (the changes are highlighted in yellow). The responses to the comments are given below.

Reviewer:

1. In introduction Line 90. There is a statement that .."[...] They are also regarded as promising antibiofilm agents [39,40].", but the references are connected with antiiofilm activity of antiseptics. In my opinion it woudl be good to enrich this part in some information about anibiofilm capabilities of AMPs. ...

Authors: 

We thank the Reviewer for this important remark. Somehow we have forgotten to mention AMPs potential against biofilms in the introduction section. Thank you for catching us up on this oversight, we included the information and the links into the introduction paragraph discussing the AMPs benefits and drawbacks as potential therapeutics (page 2, yellow highlighting).

 Reviewer:

2. Line 157. Methods for examing antibacterial effects.

There is no need to add information on percentage (2,1%) of MHB. Just MHB, as it well descrbed in literature and in broth-microdilution protocols.

Authors:

Thank you for the suggestion; we have omitted the percentage in all places except the first mention (just to stay on the safe side regarding the clarity of the protocol).

Reviewer:

3. Why you did not consider evaluation of antibiofilm activity by determination of minimal biofilm eradication concentration (MBEC) values alone and/or in combination (since you have used resazruin and or crystal violet in your assays...

In my opinion not alwas sub-inhibitory concentrations will give insights into antibiofilm potential. It must be taken into account that biofilm is a multilayered formation and peptides/antiseptics do not penetrate and disrupt biofilm in the linear manner. This should be discussed. 

Authors:

We thank the reviewer for this important comment. Though, we briefly covered the full biofilm formation inhibition (around planctonic MICs) alongside the partial sub-MIC effects, the eradication of the pre-formed biofilms usually require the use of the quite substantial supra-MIC concentrations of AMPs. We planned to narrow down the set of combinations first, and approach the question of activity against pre-formed biofilms in later publications, when we will be aiming more closely to the practical development, rather than at this starting survey phase, where we were driven by kind of a semifundamental interest. At the same time, we completely agree with your concern regarding the whole lot of a difference that may lay between the prevention of the biofilm formation and an actual disruption of the mature biofilm, so we tried our best to address the limitations of our current antibiofilm testing in the section we discuss its results (3.6, yellow highlighting).

Reviewer:

In my opinion presentation of the results included  in Tables: 2,4,5, and 6 would be more elegant, when presented on heat maps, for instance please use: https://software.broadinstitute.org/morpheus/

Authors:

It is actually a great visualization idea. Though, as we were defining synergy/additivity/antagonism by certain ranges of fractional indices’ values, we decided that omitting the numbers completely can be a bit disorienting and just hybridized our tables with the Morpheus’ generated heat maps. We also want to express our gratitude to you for pointing us to this useful online tool we were not previously aware about. It will be of a great help in our work far beyond the scope of the current research.

We greatly appreciate your input into our study!

Reviewer 2 Report

The manuscript is written well but I have some suggestion may be those will help to improve the soundness and novelty of this paper.

  1. Most antimicrobial peptides are not very dangerous, but most antiseptics are dangerous chemicals. The relationship between the two is explained in the introduction.
  2. Describe the characteristics and features of an antimicrobial peptide and their demerits.
  3. In the introduction, I gave examples and references of the different natural places where the antimicrobial peptide comes from and how it is made.
  4. In section "Bacterial Strains," claim the drug-resistant resistance in Klebsiella pneumoniae ESBL 344/17, Staphylococcus aureus 1399/17, and Acinetobacter baumannii 7226/16 and add a description.
  5. The authors selected the coli model for testing mechanisms, while antibacterial activity was conducted on Klebsiella pneumoniae ESBL 344/17, Staphylococcus aureus 1399/17, and Acinetobacter baumannii 7226/16. Why not Staphylococcus aureus?
  6. As the authors explain, is drug transport or the mechanism lac operon-dependent? Add the description with suitable justification.
  7. The resazurin assay is a fluorometric assay used to detect viable cells, but previous researchers used it to detect metabolites.Add its detection principle with suitable references.
  8. A positive control is needed to conclude the significance of the antimicrobial peptide-antiseptic combination.
  9. Because AMPs are sensitive to temperature, pH, salt, and surfactants, you can say that they store better.
  10. How the study is better and cheaper than a mix of antibiotics, antiseptics, or antiseptics that use natural extracts
  11. Refine the conclusion and abstract, indicating the clear use of combination as an antiseptic, especially for topical application, or as another form of formulation.

Author Response

We thank the reviewer for the even assessment of the manuscript. The manuscript has been corrected according to the Reviewer's comments (the changes are highlighted in green). The responses to the comments are given below.

Reviewer:

1. Most antimicrobial peptides are not very dangerous, but most antiseptics are dangerous chemicals. The relationship between the two is explained in the introduction.

2. Describe the characteristics and features of an antimicrobial peptide and their demerits.

3. In the introduction, I gave examples and references of the different natural places where the antimicrobial peptide comes from and how it is made.

Authors:

Thank you for the suggestions. We elaborated our introduction regarding the paragraphs 1-3 (green highlighting).

Reviewer:

4. In section "Bacterial Strains," claim the drug-resistant resistance in Klebsiella pneumoniae ESBL 344/17, Staphylococcus aureus 1399/17, and Acinetobacter baumannii 7226/16 and add a description.

Authors:

We thank the reviewer for drawing attention to an unclear point in the section “Bacterial strains”. Probably, we were not accurate with our phrasing regarding the description of the bacterial isolates; we clarified this now and highlighted where the link can be found. The reference we provided in the section “Bacterial strains” supplies previously published data on the MIC values of the antibiotics against all of the tested isolates, not only E.coli ESBL 521/17. As the cited paper with detailed drug-resistance data of the utilized strains is an open source, we think that including it again into the current article will be an unjustified republishing of the same material.

Reviewer:

5. The authors selected the coli model for testing mechanisms, while antibacterial activity was conducted on Klebsiella pneumoniae ESBL 344/17, Staphylococcus aureus 1399/17, and Acinetobacter baumannii 7226/16. Why not Staphylococcus aureus?

6. As the authors explain, is drug transport or the mechanism lac operon-dependent? Add the description with suitable justification.

Authors:

The explanations for the paragraphs (5) and (6) are linked.

For (6) the answer is that it is the detection of the membrane damage that is dependent on the lac operon modifications of the E. coli ML-35p strain. We added a clarification into the description of the method (section 2.2.3; green highlighting). The transport or mechanism of bactericidal action of the tested antiseptics or AMPs is not dependent on the lac operon functioning to the best of our knowledge.

The (6) partially explain the (5). The used strain (E. coli ML-35p) cannot be easily substituted by other bacteria within the scope of the method we used for analyzing membrane damaging kinetics. The method is a well established one and is routinely used in AMPs studies. Its benefits are stable reproducible results, the possibility to separately analyze the state of the inner and outer membranes of the gram-negative bacteria, and quick and frequent detection of changes, that is important for studying substances disrupting the membranes within minutes. Also, the detected permeability suggests the formation of the more serious gapes in the membranes, because the chromogenic substrates are quite large molecules, compared with some other quick and sensitive methods such as ion (e.g. K+) release from the cells that can be detected using microelectrodes. All in all, as most of the strains we analyzed in the current study are gram-negative as well, we think that the effects against this model E. coli should be informative enough.

Concerning Staphylococcus aureus, there actually once was a method based on the same principle, as the one for E. coli ML-35p, that used the S.aureus 710A strain (with constitutive beta-galactosidase, but impermeable for galactoside-based dyes) [https://febs.onlinelibrary.wiley.com/doi/epdf/10.1111/j.1432-1033.1997.0549a.x , https://doi.org/10.1016/S0021-9258(18)93523-4]. Unfortunately, it was less stable and had not gain enough popularity, and in the end the chromogenic substrate it required (NpGal6P) ended up being out of production. We have plans for trying to revitalize this method with, maybe, some alternative substrates to improve the stability and outmaneuver the issues with the NpGal6P purchase, as the verification of the results of the E.coli-based permeability testing on a gram-positive strain would be a valuable asset to form a whole picture. But at the moment we are far from completing this task. 

Reviewer:

7. The resazurin assay is a fluorometric assay used to detect viable cells, but previous researchers used it to detect metabolites. Add its detection principle with suitable references.

Authors:

We added the information about the detection principle into the section 2.2.4.

Reviewer:

8. A positive control is needed to conclude the significance of the antimicrobial peptide-antiseptic combination.

Authors:

Synergistic and antagonistic interactions are defined as “better than additive” and “worse than additive”. Thus, both synergy and antagonism are a range of effects rather than some particular effect that can be verified through comparing it to an etalon value. So, in synergy/antagonism testing one go with imitating or mathematically calculating the additive effect (as all other effects are defined through it) based on the individual activities of the substances, than establishing the error margin and comparing the results of the test with the estimated additive one. If the difference is more, than the error margin, synergy or antagonism are regarded as statistically significant.

Our testing was based on Loewe additivity paradigm. The basic principle for calculating the additive effect using this approach is that each substance is self additive: the effect of 1/2 of a dose of substance A + another ½ of a dose of A should be the effect of a full dose of A (= additivity). Using fractional index equation: FI= (½ A)/(A)+( ½ A)/(A)=1. So, the model index value for additivity should be 1. The error margin is based on 2-fold serial dilutions error margin (which is also 2 fold).  If the error occurred, ½ A can be actually ¼ A or A. So, the additive FI within the error margins can fluctuate from (¼ A)/(A)+( ¼ A)/(A)=0,5 to (A)/(A)+(A)/(A)=2 => that is from where the FI intervals for synergy (FI<=0.5) and antagonism (FI>2 or >4) originate from.

Back to the control problem, the value to control here is additivity, but it generally cannot be created directly, just calculated based on the effects of individual substances. So, these individual effects became the controls of a sort. That is why we stressed in section 2.4 that we verified the level of effect of the individual substances alongside the testing for combinations in each experiment.

Now we added some additional clarification into the section 2.2.2, where the fractional indices were first mentioned.

Reviewer:

9. Because AMPs are sensitive to temperature, pH, salt, and surfactants, you can say that they store better.

Authors:

We added minor remarks on the storage problems into the introduction and conclusion sections. But our current study was not actually focused on overcoming stability limitations of AMPs, and there is not much we can add on the subject.

Reviewer:

10. How the study is better and cheaper than a mix of antibiotics, antiseptics, or antiseptics that use natural extracts

Authors:

Some benefits of synergistic combinations in general are covered throughout the text (improved activity, lowered toxicity and chances of inducing resistance); however, it is difficult to evaluate which classes of compounds will guarantee better results. AMPs/antibiotics can win for universality (the prospects of general administration, but higher risks of resistance; more predictable synergy manifestation, but not as universal in spectrum, as with antiseptics), AMPs/antiseptics—for low resistance issues (and better improvement of toxicity problem, just because the problem itself is more prominent for antiseptics). We have some limited experience with analyzing triple combinations (work is in progress) AMP+antibiotic+antiseptic and comparing it with the effects in pairs AMP+antibiotic, AMP+antiseptic and antiseptic+antibiotic. Within the set of compounds we tested, the frequency of synergy occurrence between AMPs and antiseptics is higher, than between antibiotics and antiseptics. But it will be preposterous to say, if it is the case on a greater scale. Maybe it is true only for the particular set of compounds. Our colleagues that work with antiseptics more closely and were the enthusiasts behind the current project, says, that it is indeed hard to find compounds that synergize with antiseptics, but we do not have the proper statistics.

As of the cost, it can differ greatly depending on the compound structure. Modern recombinant technologies seem to be able to sustain mass production of AMPs at a reasonable cost, but it is unlikely, that it would get cheaper than for small molecules antibiotics or antiseptics. On the other hand, AMPs research and the effort on promoting them as therapeutic, as well as the whole thing with researching new approaches to overcome bacterial resistance or to stop its spread and emergence, were never about cost-effectiveness, but about necessity.

All in all, we think, properly answering the question you posed here based on facts rather than our speculations and believes requires a serious literature digging and analysis, and, probably, should be reserved for a focused review. The current study was kind of a “test of water”, so, not the best place to go too far into ifs and whens territory. However, your suggestion is a good point for us to cover and improve our knowledge about, as it will be a valuable argument in proceeding for the practical combinations-based design and justifying future funding requests for it.

Reviewer:

11. Refine the conclusion and abstract, indicating the clear use of combination as an antiseptic, especially for topical application, or as another form of formulation.

Authors:

We extended the abstract and added a conclusion to more precisely state the area where our findings can be applied in a more practical aspect. Though, a bit cautiously, as we still are more at the beginning stages (the search and survey for better variants), rather than at a clear path to develop a particular formulation.

Authors are very grateful to the Reviewer for the detailed analysis of the manuscript and important remarks that helped to improve the article.

Reviewer 3 Report

Generally, the proposed manuscript is well-prepared and deals with an important topic.  

In my opinion, there is a lack of a conclusion that briefly describes the most important findings.  

Therefore, I recommended it for publication after minor revision. 

Author Response

The authors are very grateful to the Reviewer for this positive overview!

According to the Reviewer's suggestion we included the Conclusion chapter into the manuscript  (the changes are highlighted in blue).

Round 2

Reviewer 2 Report

All of the comments have been taken into account, and I recommend the acceptance of the manuscript.

Author Response

Reviewer:

All of the comments have been taken into account, and I recommend the acceptance of the manuscript.

Authors:

Dear Reviewer,

We are grateful to you for this positive conclusion.

Thank you very much for investing your time and dealing with our manuscript, for very important and useful comments: it helped to direct our attempts to improve the manuscript.

Best regards,
